# Antioxidant Therapies in the Treatment of Multiple Sclerosis

**DOI:** 10.3390/biom14101266

**Published:** 2024-10-08

**Authors:** Félix Javier Jiménez-Jiménez, Hortensia Alonso-Navarro, Paula Salgado-Cámara, Elena García-Martín, José A. G. Agúndez

**Affiliations:** 1Section of Neurology, Hospital Universitario del Sureste, E28500 Arganda del Rey, Spain; hortalon@yahoo.es (H.A.-N.); paula.salgado@salud.madrid.org (P.S.-C.); 2University Institute of Molecular Pathology Biomarkers, Universidad de Extremadura, E10071 Cáceres, Spain; elenag@unex.es (E.G.-M.); jagundez@unex.es (J.A.G.A.)

**Keywords:** multiple sclerosis, treatment, pathogenesis, antioxidants, risk factors, oxidative stress, animal models

## Abstract

Several studies have proposed a potential role for oxidative stress in the development of multiple sclerosis (MS). For this reason, it seems tentative to think that treatment with antioxidant substances could be useful in the treatment of this disease. In this narrative review, we provide a summary of the current findings on antioxidant treatments, both in experimental models of MS, especially in experimental autoimmune encephalomyelitis (EAE) and in the cuprizone-induced demyelination model, and clinical trials in patients diagnosed with MS. Practically all the antioxidants tested in experimental models of MS have shown improvement in clinical parameters, in delaying the evolution of the disease, and in improving histological and biochemical parameters, including decreased levels of markers of inflammation and oxidative stress in the central nervous system and other tissues. Only a few clinical trials have been carried out to investigate the potential efficacy of antioxidant substances in patients with MS, most of them in the short term and involving a short series of patients, so the results of these should be considered inconclusive. In this regard, it would be desirable to design long-term, randomized, multicenter clinical trials with a long series of patients, assessing several antioxidants that have demonstrated efficacy in experimental models of MS.

## 1. Introduction

Multiple sclerosis (MS) is a chronic autoimmune disorder that affects the central nervous system (CNS). It has a genetic predisposition whose main characteristics are inflammation, as well as demyelination, and neuronal degeneration. Genome-wide association studies (GWAS) have found to date, at least 200 loci that have been significantly associated with the risk of developing MS [1,2]. However, most of these associations have shown weak odds ratios (OR) and only explain a part of MS heritability [1,2]. The locus with the strongest association with MS risk is *HLA* (especially the haplotype *HLA-DRB1*15:01*) [1]. Together with the genetic predisposition, it is likely that environmental factors, as well as gene-environment, plus environment-environment interactions, such as low sun exposure/low vitamin D levels, infections (mainly Epstein–Barr virus seropositivity or exposure), obesity, and smoking, may be linked to the etiopathogenesis of MS as well as its onset and progression [3,4,5]. Many data published in recent years suggest that oxidative stress, which is closely related to inflammation [6,7,8], probably also plays a prominent role in MS pathogenesis [9,10,11,12].

This possible role of oxidative stress in MS makes it reasonable to attempt treatments with antioxidant substances. In this narrative review, we have performed an exhaustive description of the results of studies addressing the usefulness of antioxidant therapies in experimental models of MS (mainly experimental autoimmune encephalomyelitis—EAE—and the cuprizone model of demyelination) and patients with MS. We underwent a search using the PubMed Database, in the time range 1966 up to 23 July 2024, crossing the term “multiple sclerosis” with “antioxidant therapy” and “antioxidant treatment”. We retrieved a total of 1566 references, which were examined manually one by one, selecting only those strictly related to this issue (finally, 283 references).

## 2. Antioxidant Therapies Tested in Experimental Models of or in Patients Diagnosed with Multiple Sclerosis

### 2.1. Alpha-Lipoic (Thioctic) Acid

Alpha-lipoic or thioctic acid (ALA, Figure 1) is a caprylic acid derivative. It is synthesized in the mitochondria (acting as a cofactor in the enzymatic nutrient breakdown) and is available in several foods. ALA contains a dithiol functional group that neutralizes reactive oxygen species (ROS) like superoxide, hydroxyl radicals, and singlet oxygen and also functions as an iron chelator [13]. Oral administration of ALA in humans showed good pharmacokinetic parameters, being their plasmatic levels comparable to those obtained in mice after administration of high doses subcutaneously [14].

#### 2.1.1. Studies in Experimental Animal Models

Intraperitoneal [15] or subcutaneous [16] administration of ALA has shown a neuroprotective action in the prevention of proteolipid protein (PLP) 139-151 peptide [16] or myelin oligodendrocyte glycoprotein 35-55 (MOG35-55)-induced experimental autoimmune encephalomyelitis (EAE). This effect is related to reducing demyelination and inflammation [15], inhibiting T-cell migration [16,17], and reducing the expression of intracellular and cell adhesion molecules-1 (ICAM-1 and VCAM-1) [17]. ALA inhibited the upregulation of ICAM-1 and VCAM-1 in murine brain endothelial cell line cultures as well [17].

Subcutaneous ALA administrated to a rodent model of EAE prevented, in a dose-dependent manner, the development of clinical signs of EAE. This prevention is caused by decreasing monocyte infiltration into the CNS, reducing monocyte migration across the blood–brain barrier (BBB), stabilizing the BBB, and reducing radical oxygen species production [18].

ALA administered subcutaneously to rodents with EAE also induced endogenous (peroxisome-proliferator-activated receptor-γ (PPAR-γ)) centrally and peripherally, suppressed encephalitogenic Th1 and Th17 cells and increased splenic regulatory Treg-cells, resulting in improvement of clinical signs of EAE [19]. ALA significantly decreased CD4+ and galectin-3+ immune cells in the brain, reducing inflammation as well [20].

Intraperitoneal administration of ALA in rodents with long-term relapsing-remitting EAE was associated with milder clinical signs, increased myelin basic protein (MBP) and reduced β-amyloid peptide precursors (β-APP) expression, downregulated tumor necrosis factor- α (TNF-α) and upregulated transforming growth factor- β (TGF-β) levels, decreased malonyl dialdehyde (MDA), increased superoxide-dismutase (SOD) levels, and increased Treg levels [21].

ALA was able to reduce depression- and anxiety-like behaviors in an EAE rodent model of progressive MS [22] and improved central neuropathic pain associated with MS in EAE rodents [23]

Dietrich et al. [24] described a neuroprotective effect preventing vision loss and degeneration of the inner retinal layers in an EAE-optic neuritis model by early administration of ALA.

In the model of demyelination induced by cuprizone in rodents, ALA treatment increased the population of mature oligodendrocytes (MOG+ cells) and decreased markers of oxidative stress (reactive oxygen species (ROS), cyclooxygenase-2 (COX-2), prostaglandinE2 (PGE2), apoptosis (caspase-3) and Bcl2 associated X apoptosis regulator/Bcl2 apoptosis regulator—Bax/Bcl2—ratio) in the corpus callosum, reducing its demyelination [25]. In the same model, ALA showed remyelinating activities, which were associated with upregulation of the expression of MBP and PLP, downregulation of the expression of TNF-α and matrix metalloproteinase-9 (MMP-9), and decrement of serum interferon-γ (IFN-γ) levels [26].

In summary, an important number of studies has shown a beneficial effect of ALA (thioctic acid), both in the treatment and prevention of the two common experimental models of MS. This improvement was due to different mechanisms related to the prevention of oxidative stress, neuroinflammation, and apoptosis [15,16,17,18,19,20,21,22,23,24,25,26].

#### 2.1.2. Studies in Human Cell Cultures

ALA and its reduced form can inhibit T-cell migration and reduce matrix metalloproteinase-9 (MMP-9) activity in a dose-dependent fashion in human T-cell line cultures, suggesting a neuroprotective action in this model [27]. Treatment with ALA also downmodulated CD4 expression in a concentration-dependent manner in cultures from human peripheral blood mononuclear cells (PBMC) as well as T cell lines [28] and stimulated the production of cAMP by G protein-coupled receptor-dependent and independent mechanisms in PBMC [29]. Treatment with ALA reduced monocyte-enriched PBMC migration both in patients with relapsing-remitting MS (RRMS) and healthy controls (basal migration being higher in RRMS than in controls) [30] and inhibited monocyte secretion of cytokines such as the interleukins IL-6 and IL-1β, and TNF-α [31].

#### 2.1.3. Studies in Patients with Multiple Sclerosis

In 1963, Mattman et al. [32] reported, in an open-label study involving 14 patients diagnosed with MS, complete or near-complete remission of symptoms in 6 patients, a slight improvement in 6, and no change in 2 of them, after administration of thioctic acid. This study was the first attempt at treatment of MS patients with an antioxidant substance.

In a double-blind, placebo-controlled study using three different regimens of ALA involving 37 MS patients, Yadav et al. [33] showed good tolerability and a dose-dependent decrease in serum levels of the soluble intercellular adhesion molecule-1 (sICAM-1, and MMP-9).

A double-blind, randomized, placebo-controlled clinical trial involving 52 patients with RRMS showed decreased plasma total antioxidant capacity (TAC), IFN-γ, ICAM-1, TGF-β and IL-4 in patients under ALA, while other markers of oxidative stress (malondialdehyde (MDA) levels, SOD and glutathione peroxidase (GPx) activities), markers of inflammation (TNF-α, IL-6, and MMP-9), and Expanded Disability Status Scale (EDSS) did not change significantly [34,35].

Another double-blinded placebo-controlled clinical trial involving 24 RRMS patients showed decreased levels of asymmetric dimethylarginine (ADMA, a precursor of nitric oxide) and a lack of increase of EDSS in patients under ALA therapy [36]. Treatment with ALA induced an increase in cyclic adenosine monophosphate (cAMP) levels at 2 and 4 h in 21 patients with secondary progressive MS (SPMS) and in 20 healthy controls, while induced a decrease of this value in 26 RRMS patients [37].

In summary, the clinical improvement of MS symptoms with ALA described in a first open-label study [32] has not been confirmed in several double-blind placebo-controlled studies [33,34,35,36,37] despite ALA being able to decrease several biochemical parameters related to oxidative stress and inflammation.

### 2.2. Melatonin

Melatonin, also known as (*N*-[2-(5-methoxy-1H-indol-3-yl) ethyl] acetamide or 5-Methoxy-*N*-acetyl-tryptamine, Figure 2), is an indoleamine derivative. It is secreted by the pineal gland, and it acts as a hormone with important functions in regulating sleep. It is also produced by some organisms, including bacteria, and is a powerful antioxidant, with both direct (neutralizing free radicals) and indirect action (increasing the expression of genes for antioxidant enzymes such as *GPx*, *Glutathione reductase*, *SOD*, and *catalase* (*CAT*)).

#### 2.2.1. Studies in Experimental Animal Models

Melatonin administered intraperitoneally showed neuroprotective effects (including improvement of clinical severity and reducing the number of demyelinating plaques and lymphocytic infiltration) in rodent models of EAE by several mechanisms:Decreasing peripheral and central T helper1/T helper 17 lymphocytes (Th1/Th17) responses and increasing the T regulatory (Treg) frequency and the synthesis of IL-10 in the Central Nervous System (CNS), therefore reducing the pro-inflammatory response [38];Decreasing the levels of oxidative stress markers (decreased thiobarbituric acid reactive substances (TBARS) and ROS concentrations and increased the level of SOD and CAT in the brain) by activation of the transcription factor NF-E2 related factor (Nrf2) and antioxidant response elements (ARE) pathway, increasing the expression of the enzymes heme oxygenase-1 (HO-1) and nicotine adenine dinucleotide(phosphate) (NAD(P)H dehydrogenase [quinone] 1 (NQO1)) [39];Reversing the decrease in glutathione (GSH) partially, the increase in oxidized glutathione (GSSG), the decrease in GSH/GSSG ratio, the decrease in GPx, and the increase in lipoperoxides, nitric oxide (NO) metabolites, carbonylated proteins, and TNF-α, caused by the induction of EAE [40];Reducing the mRNA expression of several kynurenin regulatory enzymes (mainly indoleamine 2,3-dioxygenase 1 or IDO-1) and aryl hydrocarbon receptor (AhR) and inhibiting the enzyme Nicotinamide *N*-Methyltransferase (Nnmt) overexpression (which leads to an increase in NAD+ levels) [41].

Administration of intraperitoneal melatonin to rodents with EAE, alone or associated with IFN-β1b or glatiramer acetate, improved clinical scores and reversed the increase in NO metabolites, MDA, and 4-hydroxyalkenals (end products of lipid peroxidation) in the brain, and the increase of inflammatory markers such as TNF-α, IL-1 β, and IL-6 in plasma [42].

Subcutaneous administration of melatonin to EAE mice, compared to vehicle, also improved clinical severity, decreased infiltrating leukocytes and the percentage of Th17 cells in the CNS, reduced T cell proliferation and the percentage of CD19+ B lymphocytes in the spleen, diminished IFN-γ and IL-4 expression in the spinal cords and the expression of IL-17 in the brain and the spinal cord, and increased the IL-10 and IL-27 production in the spleen [43].

In contrast, Ghareghani et al. [44] described that the administration of melatonin orally to rats with EAE exacerbated neurological symptoms and increased serum IFN-γ and lactate levels and IFN-γ/IL-4 ratio (a marker of Th-1/Th-2), and caused an increase in lymphocyte infiltration, activated astrocytes, and demyelinated plaques in the lumbar spinal cord. The group also reported that melatonin may hinder remyelination by enhancing the inhibitory effects of brain pyruvate dehydrogenase kinase-4 (PDK-4) on the pyruvate dehydrogenase complex (PDC), a crucial enzyme in fatty acid (FA) synthesis [45]. Moreover, they showed that, in comparison with melatonin alone, coadministration of a PDK-4 inhibitor with melatonin reduced EAE disability scores, inhibited pro-inflammatory and increased anti-inflammatory cytokines, decreased expression of oligodendrocytic markers in EAE, reduced lactate levels, increased *N*-acetyl aspartate (NAA) and 3-hydroxy-3-methyl-glutaryl-coenzyme-A reductase (HMGCR), and restored PDC function [45].

Melatonin administered intraperitoneally to rodents with demyelination induced by cuprizone improved locomotor activity, increased antioxidant levels (CAT, SOD, GPx, and GSH), and reduced levels of MDA and inflammatory factors (TNF-α, IL-1β) [46].

In synthesis, the results of most studies [38,39,40,41,42,43,46], except for the opposite results found by one research group [44,45], have shown a beneficial effect of melatonin in experimental models of MS, including improvement in both clinical parameters and markers of inflammation and oxidative stress.

#### 2.2.2. Studies in Human Cell Cultures

It has been reported that administration of melatonin to PBMC cultures from patients with MS and controls caused an increase in messenger ribonucleic acid (mRNA) expression and activities of CAT and sirtuin-1 (SIRT-1) in both MS patients and controls and increased mRNA expression and activity of SOD-2 only in MS patients [47]. In addition, it can decrease Th1 (with a decrease of IL-2, IL-12, IFN-γ, and TNF-α), Th19, and Th22 responses in RRMS patients (but not in controls) without affecting Th17 and Treg subsets and IL-10, IL-17, and IL-22 production, and causes an increase in anti-inflammatory/Th1 ratio, and overexpression of the melatonin effector/receptor system [48].

#### 2.2.3. Studies in Patients with Multiple Sclerosis

One study in patients diagnosed with SPMS has shown that the administration of melatonin caused a decrease in MDA concentrations and an increase in SOD and GPx activities (without significant changes in catalase activity) [49].

Adamczych-Sowa’s group has shown that administration of melatonin to patients with RRMS decreased serum total antioxidant status (TOS) [50], serum ceruloplasmin levels [51], and plasma lipid hydroperoxide levels [52], and increased serum SOD activity [53], and improved sleep quality [50] and fatigue score [52], without affecting the EDSS [50,51,53].

A double-blind, randomized, placebo-controlled clinical trial with a parallel-group design involving 26 MS patients treated with interferon beta (13 of them receiving a supplement of melatonin and the other 13 receiving placebo) with 1-year of follow-up showed a lack of beneficial effect of melatonin on the number of relapses, EDSS, appearance of new lesions in magnetic resonance imaging (MRI), fatigue and depression, although patients under melatonin showed a trend towards improvement in the performance of Multiple Sclerosis Functional Scale [54]. Another 6-month, double-blind, randomized, placebo-controlled trial involving 36 RRMS treated with IFN-β (13 of them receiving a supplement of melatonin and the other 13 placebos) showed a lack of changes in EDSS and depression scores but a significant decrease in serum concentration of oxidative stress markers (lipoperoxides and NO catabolites) and pro-inflammatory cytokines, such as TNF-α, IL-6 and IL-1β, in subjects treated with melatonin in comparison with those treated with placebo [55].

In summary, even though melatonin treatment has shown a decrease in markers of oxidative stress [49,50,51,52,53,55] and inflammation [55] and the improvement of some specific symptoms such as fatigue [52] or sleep quality [50], it has not shown improvement in global clinical parameters [50,51,53,54,55] or radiological parameters [54].

### 2.3. Epigallocatechin-3-Gallate (EGCG, Green Tea)

EGCG, a type of catechin or polyphenol, which is a major component of green and white tea, is an ester of epigallocatechin and gallic acid (Figure 3) that acts as a powerful antioxidant but also as an antiangiogenic and antitumor agent and as a modulator of tumor cell response to chemotherapy [56].

#### 2.3.1. Studies in Experimental Animal Models

Oral administration of EGCG during induction of EAE in mice reduced the clinical severity of EAE [57,58], reduced proliferation and production of TNF-α by encephalitogenic T cells, downregulated the cyclin-dependent kinase 4, and blocked the activity of the 20S/26S proteasome complex (these resulted finally in inhibition of nuclear factor-kappaB -NF-κB- activation), and protected against neuronal injury induced by *N*-methyl-D-aspartate (NMDA) in living brain tissue, and blocking the formation of neurotoxic reactive oxygen species in neurons [59]. In addition, oral EGCG reduced the production of IFN-γ, IL-6, IL-17, and IL-1β, as well as TNF-α, and also decreased Th1 and Th17 helper cells, and increased regulatory T-cell populations in lymph nodes, the spleen, and the central nervous system, and the plasma levels of intercellular adhesion molecule 1, as well as Chemokine receptor 6 (CCR6) expression in CD4(+) T cells [57].

Intraperitoneal administration of EGCG in a model of mice with EAE reduced EAE severity and macrophage inflammation in the CNS and suppressed M1 macrophage-mediated inflammation in the spleen, being these effects related to the inhibition of the signaling by NF-κB and glycolysis in macrophages by EGCG in macrophages [58].

Intraperitoneal administration of EGCG in mice with cuprizone-induced demyelination showed a significant increase in expression in the cerebral cortex of PLP and oligodendrocyte transcription factor 1 (Olig1) compared to those receiving phosphate-buffered saline or those without injection, suggesting a neuroprotective action [60].

Overall, the results of studies using EGCG showed clinical and/or inflammatory marker improvement in experimental models of MS [57,58,59,60].

#### 2.3.2. Studies in Patients with MS

A phase I single group 6-month futility study involving 10 patients with RRMS or SPMS under glatiramer acetate showed abnormal liver function tests in one patient and a 10% increase of *N*-acetyl aspartate levels after administering EGCG. This study was followed by a 12-month randomized, double-blind, placebo-controlled study with a parallel-group design involving 12 RRMS or SMPS patients under glatiramer acetate or INF-β that needed to be stopped because 5 out of 7 patients treated with a different lot of EGCG showed abnormal liver function tests [61].

A phase II randomized, double-blind, parallel-group trial evaluating oral EGCG (up to 1200 mg daily) or placebo over 36 months, with an optional 12-month open-label extension for EGCG treatment involving 61 patients with PPMS or SPMS (31 assigned to placebo). The trial was completed by 19 patients per group (in the EGCG group, two patients did not tolerate medication, and another showed abnormal liver function tests). In comparison to placebo, EGCG treatment showed a lack of efficacy in preventing brain atrophy or in changing the number and volume of lesions and in the number of contrast-enhancing lesions in MRI T2w images, changes in EDSS, annualized relapse rate, and other clinical parameters [62]. Another trial involving 122 RRMS under glatiramer acetate (60 assigned to placebo) showed that, compared to placebo, the administration of EGCG 800 mg daily during a period of 18 months did not improve either radiological or clinical parameters [63].

In contrast, a pilot randomized clinical trial involving 51 patients with MS (37 RRMS and 14 SPMS), 27 of them receiving 800 mg of EGCG and 60 mL of coconut oil, and 24 receiving a placebo for 4 months, showed improvement in the treated group in gait and balance assessed by specific scales [64].

Finally, a crossover trial at a clinical research center involving eight MS patients treated with glatiramer acetate, with 4 weeks of washout between the two treatments, showed that, compared to placebo, EGCG (600 mg/d) decreased energy expenditure, carbohydrate oxidation, adipose tissue perfusion, and glucose supply in men and increased these parameters in women at rest, while postprandial energy expenditure during exercise decreased, suggesting an increased working efficiency [65].

According to data from studies with EGCG in patients with MS, although this drug may improve some metabolic parameters [65], it has not shown improvement in clinical or radiological parameters of the disease [62,63] (except for a slight improvement in gait and balance shown in a single study in combination with coconut oil [65]). In addition, its use is not advisable due to adverse effects [61,62].

### 2.4. Curcumin

Curcumin (Figure 4) is a polyphenolic phytochemical obtained from Curcuma longa (turmeric) that has important antioxidant properties [66].

#### 2.4.1. Studies in Experimental Animal Models

Intraperitoneal administration of curcumin to an EAE model in mice showed clinical improvement associated with partial reverse of the elevated levels of IFN-γ and IL-17 and the increased expression of IL-12 and IL-23, together with an up-regulation of IL-10, peroxisome proliferator-activated receptor γ (PPARγ) and CD4(+)CD25(+–), forkhead box p3 (Foxp3) (+) Treg cells in the central nervous system and lymphoid organs [67]. Treatment with curcumin also reduced in a dose-dependent way the secretion of IFNγ, IL-17, IL-12, and IL-23 in cultures of spleen cells from EAE mice [68]. In the same model, it has been described clinical improvement associated with a significant reduction in the expression levels of some pro-inflammatory cytokine genes (*IL-6*, *IL-17*, TNF-α, and *IFN-γ*), a significant increase in the transforming growth factor β (TGF-β), and an increase in the mRNA expression and the activity of antioxidant enzyme GPx-1, without affecting *IL-12*, *IL-4*, *IL-5*, and *CAT* genes expression [67].

Intraperitoneal administration of nanocurcumin to an EAE model in rats also showed clinical improvement associated with a decrease in demyelination and a reverse in the increase in mRNA expression for *MCP-1*, *IL-17*, *NF-κb*, and *TNF-α* receptor (pro-inflammatory genes), in the decrease in mRNA expression of *IL-4*, *IL-10*, *TGF-β*, and *FOXP3* (anti-inflammatory genes), and the increase in the mRNA expression of the *inducible nitric oxide synthase (iNOS)* (pro-oxidant gene), and to an increase in mRNA expression of the antioxidant genes *HO-1* and *Nrf2*, in the spinal cord of this model [69].

Intraperitoneal administration of curcumin in mice with cuprizone-induced demyelination showed a partial reverse of the damage of oligodendroglial lineage cells (OLLC), the decrease in myelin density and myelin basic protein, and the increase in glial fibrillary acid protein (GFAP, that is a marker of astrocyte proliferation) and the ionized calcium-binding adapter molecule 1 (Iba1, that is a marker of microglia proliferation) induced by this toxin [70]. Administration of seed oil-based nanoformulations of curcumin to Swiss albino male mice (SWR/J) with cuprizone-induced demyelination improved impairment in working memory, reversed histological changes in the hippocampus, decreased the production of ROS and increased brain levels of glutathione and antioxidant enzymes such as CAT, GPx, and SOD, and showed inhibitory effect on NFκB-p65 [71].

In another model of demyelination, induced by ethidium bromide (EB) in Wistar rats, administration of curcumin or a conjugated linoleic acid-curcumin resulted in a significant improvement in spatial memory function and reduction of oxidative stress parameters in the brain (increase in TAC, decrease in CAT and SOD activities, and a decrease in the levels of MDA) [72].

Overall, curcumin administration had a beneficial effect on experimental models of MS, causing both symptomatic improvement [67,68,69,70,71,72] and improvement of markers of inflammation [67,68,69], oxidative stress [67,69,71,72], and demyelination [69,70,71].

#### 2.4.2. Studies in Patients with MS

A prospective, single-center, double-blind, placebo-controlled study as add-on therapy involving 80 RRMS patients under subcutaneous IFN β-1a 44 mcg twice-a-week (40 assigned to curcumin and 40 to placebo, 36 and 34 completed a 12-month, and 23 and 20 a 24-month of follow-up, respectively) showed lack of efficacy of curcumin on relapses and disability progression and in the proportion of subjects free from new/enlarging T2 lesions. Despite the proportion of patients with combined unique active (CUA) lesions at month 12 was lower for patients under curcumin, the differences were not maintained at month 24 [73].

### 2.5. Resveratrol

Resveratrol (3,5,4′-trihydroxy-*trans*-stilbene, Figure 5) is a stilbenoid, a type of natural phenol, and a phytoalexin produced by several plants that has antioxidant, anti-inflammatory, immunomodulatory, glucose and lipid regulatory actions [74].

Oral administration of resveratrol decreased the clinical symptoms and inflammatory responses and induced apoptosis in the spinal cord in EAE-induced mice, and induced a significant down-regulation of cytokines and chemokines, including TNF-α, IFN-γ, IL-2, IL-9, IL-12, IL-17, as well as the macrophage inflammatory protein-1α (MIP-1α), and monocyte chemoattractant protein-1 (MCP-1) [75]. Oral resveratrol administered to EAE mice improved clinical symptoms, increase the production TNF-α, IFN-γ, and IL-17 in splenic T cells, increased the number of IL-17+ T cells, IL-17+/IL-10+ T cells, and CD4(–)IFN-γ + cells in the brain and spleen, and suppressed the expression of IL-6 and IL-12/23 p40, whereas it increased the expression macrophage IL-12 p35 and IL-23 p19 [76]. Oral administration of resveratrol improved clinical scores and showed suppression in proinflammatory cytokines (IFN-γ, TNF-α) and an increase in anti-inflammatory cytokines such as IL-4 and IL-10 in EAE mice, these effects being potentiated by the concomitant use of mesenchymal stem cells obtained from mouse bone marrow (mBM-MSC) [77].

Resveratrol administered intraperitoneally to EAE mice produced a dose-dependent decrease in EAE paralysis and blood–brain barrier function by ameliorating the loss of tight junction proteins, such as occludin, zonula occludens-1 (ZO-1), and claudin-5, repressing the increase in adhesion proteins ICAM-1 and VCAM-1, suppressing overexpression of pro-inflammatory transcripts iNOS and IL-1β, upregulating the expression of anti-inflammatory transcripts arginase 1 and IL-10 cytokine in the brain, downregulating the overexpressed NADPH oxidase 2 (NOX2) and NOX4 in the brain, and suppressing NADPH activity [78]. Oral administration of resveratrol to EAE mice decreased clinical severity, inflammation, and central nervous system immune cell infiltration by upregulation of the microRNA (miRNA) miR-124 while suppressing associated target gene and sphingosine kinase 1 (SK1) in encephalitogenic CD4+ T cells [79].

Oral administration of a derivative of resveratrol can prevent neuronal loss and axonal damage in the optic nerve and spinal cords in EAE mice, this effect being mediated by sirtuin-1 (SIRT-1) [80]. Intranasal administration of resveratrol-loaded exosomes derived from macrophages (RSV&Exo) was able to inhibit inflammatory responses in the CNS (reversing the increase of concentrations of TGF-β, IFN-γ, IL-1 β, IL-6, and IL-17 in brain and spinal cord) and peripheral system (spleen and blood), and to improve clinical evolution in a mouse model of EAE [81].

Intranasal administration of resveratrol nanoparticles to female C67 black/6 (C67BL/6) mice with EAE improved motor and visual symptoms, reduced inflammatory and demyelination changes in the optic nerves and spinal cord, and increased retina ganglion cell survival [82].

Oral administration of resveratrol in mice with cuprizone-induced demyelination enhanced motor coordination and balance reversed demyelination, reversed the increase in brain TBARS as well as the reduction in SOD activity and GSH levels in mitochondrial and postmitochondrial brain fractions, inhibited NF-κB signaling and increased oligodendrocyte transcription factor-1 (Olig1) expression [83].

In contrast with the results of other studies showing the neuroprotective effects of resveratrol, Sato et al. [84] reported that oral administration of resveratrol not only did not improve but exacerbated clinical symptoms and histological changes (inflammation and demyelination) in mice with EAE and Theiler’s murine encephalomyelitis virus-induced demyelinating disease (TMEV-IDD).

In summary, resveratrol improved clinical parameters [75,76,77,78,79,80,81,82,83,84], inflammatory markers [75,76,77,78,79,81,82,83], markers of oxidative stress [81,83], and demyelination [82,83] in most studies in experimental models of SM, except for one that found worsening of these parameters [83]. To our knowledge, no clinical trials have been published to date on the efficacy and safety of resveratrol in patients with MS.

### 2.6. Pentoxifylline

Pentoxifylline (also known as oxpentifylline, see Figure 6) is a xanthine derivative used to treat muscle pain in individuals with peripheral artery disease. It acts as a competitive nonselective phosphodiesterase inhibitor, can inhibit the synthesis of TNF-α and leukotriene (reducing inflammation and innate immunity), and has antioxidant actions [85].

#### 2.6.1. Studies in Experimental Animal Models

Okuda et al. [86] reported that oral administration of pentoxifylline did not reduce the incidence and severity of EAE in mice but, at intermediate doses, delayed the onset of this disease, reduced the mRNA levels for TNF-α, IL-1 β, and IL-6 in PBMC, and delayed the infiltration of inflammatory cells in the CNS of EAE mice [86]. Grassin et al. [87] reported a lack of efficacy of several doses of pentoxifylline in the prevention of recurrences in a model of relapsing-remitting EAE in rats. In contrast, Corrêa et al. [88] described a significant reduction of neuroinflammation in the CNS and of serum levels of IFN-γ, NO, and TNF-α, in parallel with an improvement of clinical symptoms in rats with EAE.

Administration of lisophylline, the *R* enantiomer of the pentoxifylline analog, to mice with EAE did not reduce the clinical severity of acute paralysis but decreased the number and severity of paralytic attacks with relapsing EAE; this reduction is correlated with decreased mRNA levels of IFN-γ, but not of mRNA levels of IL-12 in the spinal cord [89]. In summary, data on the clinical efficacy of pentoxyfilline of its derivatives in experimental models of MS are controversial [86,87,88,89], although most of them showed improvement of inflammatory markers [86,87,89].

#### 2.6.2. Studies in Patients with Multiple Sclerosis

Two pilot, open-label trials involving a small number of patients with RRMS have shown a lack of clinical efficacy of pentoxifylline in reducing EDSS [90,91,92] or in avoiding the progression of the disease [93]. Van Oosten et al. [90] described an increase in cerebrospinal fluid (CSF) and serum levels of the soluble vascular cell adhesion molecule 1 (sVCAM-1) and a lack of changes in CSF and serum levels of TNF-α and soluble intercellular adhesion molecules 1 and 3 (sICAM-1, sICAM-3). Pentoxifylline treatment did not reduce CSF [90,91] and serum levels [90] of soluble receptors for TNF-α (sTNF-R) and was badly tolerated in one of these trials, with a withdrawal rate of 55.6% [91]. In three other pilot studies, pentoxifylline reduced the early side effects of IFN-β [93,94,95], with this effect attributed to the avoidance of the upregulation of TNF-α and IFN-γ expression by IFN-β and the synergistic effects of these drugs on the upregulation of IL-10 expression and an increase of IL-10 in serum [95]. Overall, despite the reduction of the early side effects of IFN-β described in three studies [93,94,95], pentoxifylline has not shown clinical efficacy in the treatment of MS [90,91,92].

### 2.7. Vegetable and Animal Oils

#### 2.7.1. Studies in Experimental Animal Models

Oil extracts of *Nigella sativa* (an herbaceous plant of the family Ranunculaceae, an antioxidant and anti-inflammatory agent), administered orally to EAE rats, can decrease MDA levels in the spinal cord and the brain, to decrease NO levels in the brain and increase NO levels in the spinal cord [96].

A nanodroplet formulation of pomegranate seed oil (containing high levels of the polyunsaturated fatty acid (PUFA) punicic acid, one of the strongest natural antioxidants), administered at low doses to mice with EAE, significantly decreased the disease burden and reduced oxidation of lipids in brain and demyelination [97].

Walnut oil, another important antioxidant, which contains a high concentration of PUFA, especially alpha-linoleic, linoleic, and oleic acids, reduced disease severity and plaque formation in the brains of EAE mice, decreased the production of INF-γ and IL-17 in splenocytes (without affecting IL-10 and IL-5), and decreased serum IL-17 and increased IL-10 serum levels in the same EAE model [98].

Copaiba oil (obtained from genus *Copaifera*) can inhibit, in a dose-dependent manner, the production of hydrogen peroxide (H_2_O_2_), NO, IFN-γ TNF-α, and IL-17 in cultures of splenocytes from EAE mice [99].

Extra-virgin olive oil, oleic acid, and hydroxytyrosol, administered orally, were able to reduce the degree of lipid and protein oxidation and to increase GPx activity in a rat model of EAE, both in the brain, spinal cord, and blood [100]. Oral administration of olive leaf tea, followed by intraperitoneal injection of an extract of the olive leaf to rats with EAE, attenuated the clinical course of EAE, decreased MDA levels, upregulated antioxidant enzymes such as SOD1, SOD2, and GPx1, upregulated both overall as well as microglial SIRT1 and anti-inflammatory M2 microglia. It also downregulated the proinflammatory M1 type and preserved myelin integrity in the brainstem [101].

Oleacine (a phenolic compound obtained from virgin olive oil or from three olive leaves), administered to female C57BL/J6 mice with EAE, had the following effects [102]: (a) improvement of clinical symptoms, (b) demyelination, decrease in leukocyte infiltration, superoxide anion accumulation in CNS tissues and BBB disruption, (c) decrease of the expression of proinflammatory cytokines (IL-13, TNFα, granulocyte-macrophage colony-stimulating factor (GM-CSF), monocyte chemoattractant protein-1 (MCP-1) and IL-1β), (d) increase of the cytokine IL-10, (e) decrease of oxidative system parameters (f) upregulation of the ROS disruptor Sestrin-3, (g) prevention of the NLR family pyrin domain containing 3 (NLRP3) expression and the phosphorylation of p65-NF-κB, and (h) reduction of the synthesis of proinflammatory mediators triggered by inflammatory stimuli in BV2 cells [103]. This compound also reduced oxidative stress (protecting from EAE-induced superoxide anion, protein accumulation, and lipid oxidation products) and inflammation (by reducing both IL-1β TNF-α levels) in the colon of EAE mice [102].

Evening primrose/hemp seed oil (EPO/HSO) contains essential fatty acids with a favorable ratio of omega-6/omega-3 and antioxidant properties. Administration of EPO/HSO to mice with EAE increased the percentage of essential fatty acids in cell membranes of the spleen and blood and increased the relative expression in lymphocytes of several interleukins genes such as *IL-4*, *IL-5* and *IL-13*, and the serum level of IL-4 [104].

1,2,4-trimethoxybenzene (1,2,4-TTB), an active ingredient from essential oils that acts as a selective NLRP3 inflammasome inhibitor, decreased in immortalized murine bone marrow-derived macrophages caspase-1 activation and IL-1β secretion (iBMDMs) and primary mouse microglia cultures, and ameliorated EAE progression and demyelination after intragastric administration to mice with EAE [105].

Farnesol, a 15-carbon organic isoprenol synthesized by plants and mammals, which is present in many essential oils and has antioxidant, anti-inflammatory, and neuroprotective activities, delayed the onset and decreased severity of EAE in mice, being these effects linked to a significant decrease in spinal cord infiltration of monocytes, macrophages, dendritic cells, CD4+ T cells, along with alterations in gut microbiota composition [106].

Ginger (*Zingiber officinale*) essential oil, another potent antioxidant, used at three different doses, reduced demyelination of corpus callosum induced by cuprizone in rats by increasing the levels of *Mbp* and *Oligodendrocyte transcription factor* (*Olig2*) genes [107].

Eugenol, an allybenzene derivative present in certain essential oils such as clove, nutmeg, cinnamon, basil, and bay leaf, administered to C57BL/6 mice with EAE, led to a significant reduction in clinical symptom severity and suppressed EAE-related immune cell infiltration as well as the production of proinflammatory mediators [108].

In summary, experiments with different types of vegetable oils with antioxidant and anti-inflammatory actions have shown clinical improvement in different animal models of MS [96,97,98,99,100,101,102,103,104,105,106,107,108]. This effect was related to a decrease in oxidative stress [96,97,99,100,101,102,103] and inflammation markers [98,99,101,102,103,104,105,106,108] and/or with the prevention of demyelination [97,107].

#### 2.7.2. Studies in Patients with MS

Fish oil, which contains omega-3 PUFAs such as docosahexaenoic acid (DHA) and eicosapentaenoic acid (EPA), has important antioxidant and anti-inflammatory effects. A randomized, double-blind, placebo-controlled group involving 50 patients with MS treated with IFN-β1b showed a decrease in serum levels of IL-1β, TNF-α, IL-6, and NO metabolites, but lack of changes in serum lipoperoxide levels and in EDSS and annualized relapses rate in patients receiving fish oil compared with those receiving placebo group [109]. In contrast, another randomized, double-blind, placebo-controlled trial conducted with 50 participants RRMS patients under therapy with fingolimod showed a lack of changes in serum levels of TNF-α, IFN-γ, IL6, and IL-1β and EDSS at 12 months [110].

A randomized, double-blinded clinical trial involving 46 RRMS patients (23 assigned to oral fish oil and 23 to olive oil) showed, in both groups, a transitory decrease in the fluidity of mitochondrial membranes of platelets and an increase in the hydrolytic activity of ATP synthase in the mitochondria from platelets [111].

Finally, a randomized, double-blind clinical trial carried out in a single center with pomegranate seed oil involving 30 MS patients with a cross-over design during the first six months, followed by administration of the active product to all patients for the next six months, showed a lack of effect on EDSS, but a modest beneficial effect on the verbal testing in the initial 3-month period of active treatment [112].

Overall, administration of different animal and vegetable oils did not improve global clinical parameters in MS patients [109,110,112] despite some of them modifying several parameters related to oxidative stress [110,111] or inflammation [109].

### 2.8. Coenzyme Q_10_

Coenzyme Q_10_ (CoQ_10_, ubiquinone, Figure 7) is a biochemical cofactor (coenzyme) found in many organisms, including animals, which plays an important role in mitochondrial oxidative phosphorylation and acts as a powerful antioxidant and can modulate the expression of genes involved in inflammatory processes.

#### 2.8.1. Studies in Experimental Animal Models

Fiebiger et al. [113] showed a lack of beneficial effect (improvement in inflammation, demyelination, or axonal damage) of idebenone (a synthetic CoQ_10_ analog) in mice with EAE [113]. In contrast, Soleimani et al. [114] described an improvement in clinical symptoms and a significant decrease in brain levels of TNF-α and IL-10 and the ratio of TH1/TH2 interleukins (with no changes in IL-4 and IL-12) in C57BL/6 female adult mice with EAE. Khalilian et al. [115] showed that CoQ_10_ administration in the cuprizone mice model of MS increased MBP and Olig-1 expression, alleviated oxidative stress status induced by CPZ, and suppressed inflammatory biomarkers, suggesting a neuroprotective effect in this model. In summary, studies on the effect of CoQ_10_ or its derivatives on experimental models are insufficient and controversial [113,114,115], although some point to clinical improvement [114] and parameters related to oxidative stress [115] and/or inflammation [114,115].

#### 2.8.2. Studies in Patients with MS

A 12-week, double-blind, placebo-controlled study involving 45 RRMS patients (22 receiving 500 mg/day of CoQ_10_ and 23 assigned to placebo) showed lack of improvement in EDSS scores despite a significant increase in SOD activity and plasma TAC, decrease in MDA levels (with no significant changes in GPx activity) [116], decrease of plasma levels of the inflammatory markers TNF-α, IL-6, and MMP-9 (with no significant changes of the anti-inflammatory markers IL-4 and TGF-β) [117] in patients under this therapy. However, the same authors described improvement in fatigue and depression scores in the same patients under CoQ_10_ therapy [118].

An 8-week randomized placebo-controlled trial involving 28 MS patients with concurrent training and CoQ_10_ administration showed a lack of effect of CoQ_10_ in several parameters of functional capacity [119].

Finally, a double-blind, placebo-controlled phase I/II, adaptively designed, baseline-versus-treatment, placebo-controlled, CSF-biomarker-supported clinical trial of idebenone in 77 patients with PPMS (39 assigned to idebenone and 38 to placebo) showed lack of effect of this compound in inhibiting disability progression and in reducing CSF biomarkers of mitochondrial dysfunction (GDF15- and lactate), axonal damage (NFL), innate immunity (sCD14), retinal nerve fiber layer thinning and blood–brain barrier leakage (albumin quotient) [120].

### 2.9. Antioxidant Vitamins

This section describes data obtained from several studies on the possible protective role of alpha-tocopherol (Figure 8), vitamin A (retinol, Figure 9), and carotene derivatives in experimental models of MS or patients diagnosed with MS.

#### 2.9.1. Studies in Experimental Animal Models

Administration of α-tocopherol [121], its analog TFA-12 [122], or α-tocopherol emulsions to C57BL/6 adult female mice [121,122] or to female SJL/J mice [123] with EAE attenuated the severity and delayed the disease progression [121,122,123] by reducing inflammation and demyelination reaction in the spinal cord [121,122], decreasing the proliferation of splenocytes, and inhibiting production of IFN-γ (Th1 cytokine) [121] or other cytokines [123].

Combined therapy with vitamins A and C in female Lewis rats with EAE decreased neurological severity, and EAE disease progression caused a significant reduction in demyelination size, immune cell infiltration, inflammation, microglia, and astrocyte activation. Also, it caused decreased levels of pro-inflammatory cytokines (TNF-α, IL1β) and iNOS and increased the expression of the genes *HO-1*, *IL-10*, *MBP*, and *Nrf-2*, increased the TAC. and decreased levels of oxidative stress markers [124].

The carotenoid derivative bixin, administered to female C57BL/6 mice, improved the symptoms and pathology in EAE mice and decreased the release of inflammatory cytokines such as TNF-α, IL-6, IL-8, IL-17, and IFN-γ. It also suppressed microglial aggregation, increased the expression of the anti-inflammatory cytokine IL-10, and reduced the proportion of Th1 and Th17 cells in the CNS and the spleen. Additionally, it inhibited Thioredoxin-interacting protein (TXNIP)/NLRP3 inflammasome activity and decreased oxidative stress through the activation of nuclear factor erythroid 2-related factor 2 (NRF2) [125].

The carotenoid derivative crocine (present in crocus and gardenia), administered to C57BL/6 female mice with EAE, improved neurobehavioral deficits, suppressed the increased expression of stress genes in the endoplasmic reticulum such as *XBP-1/s*, *BiP*, *PERK*, and *CHOP*, preserved myelination and axonal density, and decreased T cell infiltration and macrophage activation in the spinal cord [126]. This carotenoid, administered to C57BL/6 male mice with demyelination induced by cuprizone, significantly improved several clinical parameters and reversed MDA increase and GPx, SOD, and TAS decrease in serum and brain tissue induced by this neurotoxin [127]. The analog of crocine crocetinate, administered to female BALB/C57 mice with EAE, improved clinical symptoms, decreased microgliosis, demyelination, and the levels of inflammatory markers IL-1β and TNF-α, reversed the altered levels of MDA and GSH and reduced PTEN-induced kinase 1 (PINK1) and Parkin protein levels in the spinal cord tissue [128].

Synthesized, the results of studies in which antioxidant vitamins such as α-tocopherol, vitamins A and C, or carotenoid derivatives were administered to experimental models of MS have shown clinical improvement [121,122,123,124,125,126,127,128], improvement in markers of oxidative stress [124,125,126,127,128] and inflammation [121,122,123,124,125,126,128], and/or prevention of demyelination [121,122,124,126,128].

#### 2.9.2. Studies in Patients with MS

A randomized study on interferon-β treatment, with a double-blind design, placebo-controlled, that involved 36 RRSS patients for 24 months (18 under a mixture of omega-3 and omega-6 PUFAs, vitamin A, vitamin E, and γ-tocopherol and 18 under “placebo” therapy containing olive oil) showed a significant improvement of some functional capacity and gait parameters in the group under vitamin therapy [129]. The administration of a mixture of selenium, vitamin C, and vitamin E for 5 weeks to 18 patients diagnosed with MS increased GPx activity five-fold [130]. The administration of vitamin E to 34 MS patients for 3 months caused a significant reduction in serum lipid peroxides levels [131].

A total 1-year placebo-controlled randomized clinical trial involving 101 patients diagnosed with RRMS showed significant improvement in the MS functional composite scale [132] and in depression and fatigue scales [133] but a lack of significant changes in EDSS, relapse rate, and brain active lesions [132], in patients under therapy with vitamin A. A double-blind, randomized trial by the same group, involving 39 RRMS patients, showed a significant decrease in the expression of *IFN-γ* and *T-bet* genes in the PBMC of patients under vitamin A therapy (suggesting a modulation of the impaired balance of Th1 and Th2 cells by this vitamin) [134].

A randomized, double-blind, placebo-controlled study involving 40 patients with MS showed a significant decrease in the serum levels of lipid peroxidation and DNA damage markers, TNF-α, and IL-17, and a significant increase in the serum TAC/TAS in the group assigned to the carotenoid crocine compared to those assigned to placebo [135]. Another 8-week, randomized, double-blinded clinical trial involving 50 MS patients showed improvement in anxiety scales and a decrease in serum hs-CRP levels, with no significant changes in serum MDA and NO levels in patients treated with crocin compared to those under placebo [136].

In summary, although according to some studies, the administration of antioxidant vitamins or derivatives of these to patients with MS can improve some clinical parameters [129,132,133,135], some markers of oxidative stress [130,131], or inflammation [134,135,136], to date, they do not seem to have demonstrated global clinical improvement [132], nor improvement in neuroimaging parameters [132].

### 2.10. Uric Acid and Bilirubin

Uric acid is a product of purine metabolism consisting of a heterocyclic compound of carbon, nitrogen, oxygen, and hydrogen (Figure 10), which is synthesized from xanthine and hypoxanthine by the enzyme xanthine oxidase and acts as a strong reducing agent (electron donor) and a powerful antioxidant.

Bilirubin is synthesized from the heme derivative biliverdin through the action of the enzyme biliverdin reductase, and is oxidized to biliverdin again, consists of an open-chain tetrapyrrole (Figure 11), and acts as a potent antioxidant.

#### 2.10.1. Studies in Experimental Animal Models

Experimental studies performed in 8- to 9-week-old female PL-SJLF1/J (PLSJL) mice and in interferon-gamma receptor knockout mice with EAE have shown improvement in clinical signs and long-term survival by the previous administration of uric acid [137], being these effects related with apoptotic cell death and blocking peroxynitrite-mediated tyrosine nitration in inflammation areas of the spinal cord, suppression of the enhanced blood-CNS barrier permeability characteristic of EAE [138], reduction of iNOS mRNA-positive cells in the peripheral blood and spinal cords [139], and prevention of inflammatory cell invasion into the CNS [140]. Administration of the uric acid precursors inosine or inosinic acid to female PL-SJLF1/J (PLSJL) mice with EAE significantly increased uric acid levels in the CNS and promoted recovery from clinical signs of EAE [141]. In summary, the administration of uric acid or its precursors in experimental models of MS leads to clinical improvement [137,141], a reduction in nitrosative stress [138,139], and inflammatory changes [140].

Administration of bilirubin can improve symptoms of EAE and alleviate oxidative damage in the spinal cord in male Lewis and female Dark Agouti rats, both previously and after EAE induction, although only administration before EAE showed a reduction of inflammation in histological examination [142]. Biliverdin reductase has also shown the ability to improve clinical and pathological signs of EAE in male Lewis rats [143]. Treatment with bilirubin suppressed EAE induction in SJL/J mice (in part by suppressing CD4+ T cells), while depletion of endogenous bilirubin dramatically exacerbated this disease [144].

#### 2.10.2. Studies in Patients with MS

An open-label study involving 11 patients with MS showed that 1-year oral administration of the precursor or uric acid inosine significantly increased serum and CSF uric acid levels, was related to evidence of improved function in three patients and no sign of relapsing disease in the remainder, and a notable decrease in lesion activity was observed in one of two patients with active lesions identified by MRI [145,146]. A 1-year double-blind, placebo-controlled, randomized, cross-over trial (starting with placebo for 6 months and then inosine for 6 months, compared to treatment with inosine for one year after baseline assessment) by the same group, involving 16 RRMS patients showed that increased serum UA levels correlated with a significant decrease in the number of gadolinium-enhanced lesions, improve in EDSS, and decrease in relapse rates [147].

In contrast, a randomized placebo-controlled clinical trial in 157 patients with RRMS compared the effects of IFN-β + inosine (N = 79) or IFN-β + placebo (N = 78) for 2 years showed a similar percentage of patients with progression of disability and time to sustained progression in both groups, suggesting a lack of additional benefit on disability of adding inosine compared with interferon beta alone [148]. Moreover, another 1-year randomized, double-blind, placebo-controlled trial involving 36 RRMS patients assigned to IFN-β1a 44 µg + inosine or to IFN-β1a 44 µg + placebo showed a lack of differences between the two groups in the percentage of patients without relapses, relapse rates, clinical and radiological activity of MS, and progression to secondary MS (SPMS) [149].

Despite preliminary data suggesting clinical and radiological improvement of MS by administration of inosine [146,147,148], two placebo-controlled studies [148,149], one of them with an important number of patients [149], did not find any additional improvement after using inosine as add-on therapy to IFN-β.

### 2.11. Nitric Oxide Synthase (NOS) Inhibitors and NO Scavengers and Precursors

Administration of the iNOS inhibitors aminoguanidine [150], an antisense oligodeoxynucleotide [151], or the NO scavenger NOX-100 [152] to SJL mice with EAE inhibited disease expression in a dose-related manner [150,151,152], reduced spinal cord inflammation, demyelination, and axonal necrosis [150], and reduced CNS inflammation and gene expression of proinflammatory cytokines and *iNOS* [151,152]. Aminoguanidine also improved clinical signs and evolution of EAE in Lewis rats [153] and 3-month-old female Sprague–Dawley rats [154] with this disease by decreasing markers of oxidative and nitrosidative stress [154]. Administration of the peroxynitrite scavengers mercaptoethylguanidine (MEG) and guanidinoethyldisulphide (GED) previously to EAE induction in PLSJL mice decreased the number of animals displaying EAE signs and delayed EAE onset but had no effect when administered after EAE induction [155]. Similarly, treatment with the iNOS inhibitor tricyclodecan-9-xyl-xanthogenate, or with the NO scavenger, 2-phenyl-4,4,5,5-tetramethylimidazoline-1-oxyl-3-oxide improved symptoms of EAE in SWXJ-14 female mice [156]. In contrast, treatment of Lewis rats with EAE with the NO precursor *N*-methyl-l-arginine acetate caused a significant prolongation and a worsening in clinical symptoms of the disease [157], and microinjections of this substance into the corpus callosum of Wistar rats caused demyelination and neuroinflammation [158].

To date, no studies regarding the possible role of NOS inhibitors and NO scavengers in patients with MS have been reported.

### 2.12. N-Acetyl-Cysteine

*N*-acetyl-cysteine is the *N*-acetyl derivative of the amino acid L-cysteine (Figure 12), and it is a precursor of glutathione. This compound has an important antioxidant action due to its thiol (sulfhydryl) group.

#### 2.12.1. Studies in Experimental Animal Models

*N*-acetyl-cysteine improved clinical signs and evolution of EAE in 3-month-old female Sprague-Dawley rats by decreasing markers of oxidative and nitrosative stress [155]. *N*-acetyl-cysteine partially suppressed, in a dose-response fashion, the production of nitrites and TNF-α in primary astrocyte cultures from SJL/J susceptible mice when infected with Theiler’s murine encephalomyelitis virus [159].

Administration of *N*-acetyl-cysteine to 6 to 8-week-old C3H.SW/C57/BL female mice with EAE caused a drastic decrease in the clinical signs, inflammation, MMP-9 activity, and protected axons from demyelination damage [160].

The administration of S-allyl cysteine to Dark Agouti rats with EAE improved clinical signs and reduced oxidative stress parameters of this disease [161].

Overall, *N*-acetyl-cysteine administration to several experimental models of MS improved clinical evolution [154,160,161] and reduced oxidative [154,159,161] and nitrosative stress [154,159], inflammation [159,160] and demyelination [160].

#### 2.12.2. Studies in Patients with MS

A randomized clinical trial involving 24 patients with MS assigned to *N*-acetyl-cysteine plus standard of care (N = 12) or standard of care only (N = 12) showed a significant improvement in scores of cognition and attention and an increase in cerebral glucose metabolism (assessed by positron emission tomography (PET)/MRI with ^18^F-fluorodeoxyglucose) in several brain regions including lateral temporal gyrus, inferior frontal gyrus, middle temporal gyrus and the caudate in the MS group treated with *N*-acetyl-cysteine [162]. Another randomized clinical trial involving 42 patients with MS assigned to *N*-acetyl-cysteine (N = 21) or to placebo (N = 21) showed improvement in anxiety scores and serum MDA levels in patients treated with *N*-acetyl-cysteine, while did not show significant changes in depression scores and serum NO levels and GSH erythrocyte concentrations [163].

Another randomized clinical trial involving 15 patients with PMS (10 assigned to *N*-acetyl-cysteine and 5 to placebo) showed non-significant differences in improvement in a fatigue scale, in the blood GSH/GSSG ratio, and in the GSH/creatine ratio in anterior and posterior cingulate cortex, insula, caudate, putamen, and thalamus by 7 Tesla (7T) MR spectroscopy, between the two groups [164].

Finally, a multi-site, randomized, double-blind, parallel-group, placebo-controlled add-on phase 2 trial involving 90 patients with PMS with EDSS 3.0–7.0 and aged 40–70 years, assigned to *N*-acetyl-cysteine 1200 mg twice-a-day or matching placebo (1:1 ratio) is currently undergoing [165].

### 2.13. Flavonoids

Flavonoids or bioflavonoids are a class of polyphenolic secondary metabolites with antioxidant and chelating properties found in plants. Several studies showed that the administration of certain flavonoids caused significant improvement in the severity of EAE and a reduction of demyelination and inflammatory cell infiltration in mice with this disease [166,167,168,169]. This improvement was related to the inhibition of IL-12 [167] and IL-17 production [169], inhibition of neural antigen-specific Th1 [166,169] a Th17 differentiation [169], decreased the expression of CLEC12A and α4 integrin on dendritic cells, and increased retention of immune cells in the periphery [167], decreased the expression of proinflammatory cytokines in M1 microglia/macrophages, inhibited activation of signal transducers and caused activator of transcription 1 (STAT1) [168], and decreased of chemokine receptor type 6 (CXCR6) + CD4 and CD8 cells [169].

The flavonoid compound licochalcone A (licoA), administered to C57Bl/6 mice with EAE, caused improvement of clinical severity, inhibition of H_2_O_2_, IL-17, IFN-γ, NO, and TNF-α production in splenocytes, as well as inhibition in peritoneal cells of IFN-γ, IL-17 and TNF-α production [170].

The administration of high doses of the isoflavone daidzein (7-hydroxy-3-(4-hydroxyphenyl)-4H-chromen-4-one), present in soybeans and other legumes, to C57BL/6 mice with EAE, reduced the extent of demyelination and disease severity, decreased IFN-γ and IL-12 secretion, increased IL-10 production, suppressed lymphocyte proliferation, and decreased cytotoxicity [171].

Treatment with the citrus flavonoid nobiletin administered to male C57BL/6 mice aged 8–10 weeks before or after induction of EAE improved clinical symptoms of the disease and reduced inflammatory response in the brain and spinal cord by inhibition of the EAE-induced increase of IL-1β, IL-6, and TNF-α activities, and increasing the IL-10, TGF-β and IFN-γ expressions [172].

The flavonol quercetin has shown anti-inflammatory activities and neural protective effects in an experimental model of EAE by inhibition of the activation of dendritic cells (an effect mediated by the Signal transducer and activator of transcription 4-STAT4) and modulating the Th17 cell differentiation in the co-culture system [166,173].

Rutin, a glycoside combining the flavonol quercetin and the disaccharide rutinose, present in citrus and other plants, administered to C57BL/6 mice with demyelination induced by cuprizone, significantly improved locomotor activity and motor coordination, improved remyelination, and attenuated cuprizone-induced oxidative stress and inflammation in the corpus callosum [174].

Some flavonoids were able to prevent clinical signs and histological, immunological, and biochemical changes (including oxidative stress) induced in the cuprizone model of demyelination in male Wistar rats [175] and in male C57BL/6 mice by inducing the expression of Nrf2/HO-1 and inhibiting the expression of TLR4/NF-κB [176].

Overall, many studies with different types of flavonoid derivatives have shown a protective action in experimental models of MS, including clinical improvement and a decrease in inflammation, oxidative stress, and demyelination [166,167,168,169,170,171,172,173,174,175,176].

Karpov et al. [177] described a decrease in the severity of neurological and visual symptoms in an open-label study involving 41 patients with MS (22 assigned to methylprednisolone and 19 to methylprednisolone associated with cytoflavin) in those using cytoflavin.

### 2.14. Peroxisome Proliferation Activator Receptor (PPAR)-Gamma Agonists

Administration of PPAR-gamma agonists caused clinical improvement (delayed onset and decrease of severity of clinical signs) of EAE induced in Vβ8.2 T cell receptor (TCR)-transgenic mice [178], and inhibited the production of nitric oxide, the pro-inflammatory cytokines TNF-α, IL-1β, and IL-6, and the chemokine MCP-1 from primary murine microglia and astrocytes cultures [179] and in lipopolysaccharide-stimulated microglia [180,181]. Paintlia et al. [182] described the inhibition of proinflammatory cytokines-induced NF-κB transactivation in CNS glial cell cultures by IL-4 via PPAR-gamma activation, suggesting its implication for the protection of differentiating oligodendrocyte precursors during MS and other CNS demyelinating diseases.

### 2.15. Carnitine and Carnosine

#### 2.15.1. Studies in Experimental Animal Models

Carnitine (3-hydroxy-4-trimethylaminobutyrate, L-carnitine or levocarnitine, Figure 13), quaternary amine synthesized in the liver, kidneys, and brain from lysine and methionine, transports fatty acids into the mitochondria and acts as a potent antioxidant acting as a free radical scavenger. Carnosine (beta-alanyl-L-histidine, Figure 14) is a dipeptide composed of histidine and beta-alanine, which is highly concentrated in the brain and muscles and acts as a free radical scavenger and as a transition metal sequestering agent.

Combined therapy with acetyl-l-carnitine and dexamethasone improved the clinical outcome of female Sprague–Dawley rats with EAE decreased MDA and caspase-3 levels, increased GSH levels, and Bcl-2 expression, and decreased CD4+ T cells expression in the brain and the spinal cord [183]. A study in carnitin-palmitoyl transferase 1 (Cpt1) P479L mouse strains showed a reduced susceptibility for EAE in knockout than in wild-type mice (in addition, the former increased oxidative stress markers such as Ho-1 andNox2 while the latter showed increased expression of mitochondrial antioxidants regulator Pgc1α) [184].

Administration of L-carnitine in cuprizone-induced demyelination in male Sprague-Dawley rats model improved the reduction in nerve conduction velocity, the demyelination in the sciatic nerve fibers, and the increase in IL-17 level and IL-1β, p53, iNOS, and NF-KB expression [185].

Administration of carnosine to female 9-to-11-week-old C57BL/6 OlaHSD mice with EAE increased spinal cord carnosine levels and carnosine-acrolein quenching, suppressed inflammatory activity, caused a reduction in the acrolein adduct formation, and diminished the clinical severity of the disease [186].

#### 2.15.2. Studies in Patients with Multiple Sclerosis

The administration of L-carnosine to three patients with RRMS for 8 weeks caused improvement in some clinical parameters, an increase in serum TAC, and increased brain choline-contained compounds, total creatine, and myo-inositol levels in girus cinguli assessed by single-voxel 1.5 T MR spectroscopy [187]. A 12-week, open-label study involving 28 patients with RRMS showed that a guarana, selenium, and L-carnitine-based multi-supplement, mixed in cappuccino-type coffee, was able to decrease the plasma levels of oxidized DNA and pro-inflammatory cytokines [188].

### 2.16. Edaravone

Edaravone (Figure 15), a potent antioxidant acting as a free radical scavenger, improved EAE clinical symptoms and reduced infiltration of lymphocytes and the expression of iNOS in the spinal cords in female SJL mice of 8 weeks old [189]. Similarly, it improved EAE symptoms, promoted remyelination, and increased the expression of *Mbp*, *Mog*, and *Olig2* genes in the cuprizone model of demyelination in female Wistar rats [190]. In in vitro studies, edaravone inhibited free radical production in phagocytosis of opsonized particles and protein kinase C-stimulated granulocytes from MS patients and healthy controls [191].

### 2.17. Phycocyanine/Phycocyanobiline

These phycobiliproteins, which are accessory pigments to chlorophyll in plants, have antioxidant and anti-inflammatory actions. Phycocyanine has shown the ability to prevent or downgrade EAE expression in the EAE model in male Lewis rats, 6–8 weeks old [192], and in C57BL/6 female mice, 6–8 weeks old [193]. In addition, it can reduce the downregulation of IL-1,7, the inflammatory infiltrates in the spinal cord tissue, the axonal preservation, and the expression in brain tissue and serum to improve the redox status in the later EAE model [193] and induce a regulatory T cell (Treg) response, in PBMC from MS patients [192].

Phycocyanobiline decreased pro-inflammatory cytokines (IL-17, IFN-γ, and IL-6) and markers related to apoptosis in the brain [194] and the spinal cord [195], and reduced demyelination, active microglia/macrophages density, and axonal damage and increased oligodendrocyte precursor cells and mature oligodendrocytes in the spinal cord of 8–10 week-old female C57BL/6 mice with EAE [195].

### 2.18. Antidiabetic Drugs

Liraglutide (glucagon-like peptide-1) administration to female Lewis rats with EAE significantly delayed disease onset, increased MnSOD brain levels, and reduced the APP levels in the brain, with no changes in GFAP levels [196].

Administration of metformin to C57BL/6J mice with cuprizone-induced demyelination improved motor dysfunction, increased the renewal of mature oligodendrocytes in the corpus callosum via AMPK/mTOR pathway, increased the antioxidant response in mature oligodendrocytes (Nrf2+ cells), and reduced brain apoptosis markers [197].

### 2.19. Methallothioneine

Administration of zinc-metallothionein-II (Zn-MT-II) to female Lewis rats with EAE significantly decreased the clinical symptoms, mortality, leukocyte infiltration, and the CNS expression of TNF-α and IL-6 [198].

### 2.20. Caffeic Acid

Administration of caffeic acid phenethyl ester (CAPE) to female Wistar rats with EAE improved clinical symptoms of this disease and significantly decreased oxidative stress markers such as MDA, NO, GPx, and SOD activities in the brain and the spinal cord [199]. Similarly, in female C57BL/6 mice with EAE, CAPE pretreatment decreased microglia/macrophage activation, demyelination injury, inflammatory cell infiltration, and reduced the level of Th1 cells in the CNS and the spleen, whereas it increased regulatory T cells (Tregs) in the CNS [200].

### 2.21. Histone Deacetylase (HDAC) Inhibitors

Inhibitors of histone deacetylase administered to 6–8-week female C57BL/6 mice with EAE caused improvement in clinical disability [201,202] and a significant decrease in spinal cord inflammation, demyelination, neuronal and axonal loss [201,202]. These actions were related to the upregulation of antioxidant, anti-excitotoxicity and pro-neuronal growth and differentiation mRNAs, inhibition of caspase activation, and downregulation of gene targets of the pro-apoptotic E2F transcription factor pathway [201]. In addition, they can suppress the activation of M1 microglia and the proinflammatory cytokine expression, decrease NO and iNOS levels, inhibit activation of the toll-like receptors/myeloid differentiation primary response protein 88 (TLR2/MyD88) signaling pathway, downregulate the expression of HDAC3, and upregulate the acetylated NF-κB p65 levels [202].

### 2.22. Other Antioxidants

Many other antioxidant molecules have been tested in experimental models of MS [203,204,205,206,207,208,209,210,211,212,213,214,215,216,217,218,219,220,221,222,223,224,225,226,227,228,229,230,231,232,233,234,235,236,237,238,239,240,241,242,243,244,245,246,247,248,249,250,251,252,253,254,255,256,257,258,259,260,261,262,263,264,265,266,267,268,269,270,271,272,273,274,275,276,277,278,279,280,281,282,283,284], and only a few in generally low sample size series of patients diagnosed with MS [285,286,287,288,289,290,291,292], the main results of which have been summarized, respectively, in Table 1 and Table 2.

## 3. Discussion, Conclusions, and Future Directions

Most of the published studies on the possible usefulness of treatment with various antioxidant treatments in MS have been conducted with experimental models of this disease, mainly rodents with EAE or with cuprizone-induced demyelination. However, it should be borne in mind that the results obtained in these models may not reflect those expected in patients with MS.

Numerous studies carried out with ALA, melatonin, EGCG, curcumin, resveratrol, pentoxifylline, vegetable and animal oils, CoQ10, antioxidant vitamins, uric acid, bilirubin, NOS inhibitors, scavengers and precursors of NO, *N*-acetyl-cysteine, flavonoid compounds, and other molecules with antioxidant action have shown significant improvement, ability to prevent or delay the development of symptoms of EAE, and improvement of the histopathological changes of this disease (inflammation, demyelination, axonal damage, etc.). Most of these antioxidant molecules have also shown the ability to reduce levels of pro-inflammatory cytokines in the brain, spinal cord, and peripheral tissues, increase levels of anti-inflammatory cytokines, decrease levels of oxidative stress markers, and increase those of antioxidant enzymes.

Clinical trials with ALA in patients with MS, all of them with low sample sizes [32,33,34,35,36,37] and some of them double-blind and placebo-controlled [33,34,35,36], have shown, in general, some improvement of parameters related to oxidative stress [32,33,34,35,36,37], but the only one that assessed clinical data did not show improvement in EDSS [36].

Melatonin has been tested in seven clinical trials involving patients with MS [50,51,52,53,54,55,56], with only two double-blind placebo-controlled trials with a follow-up period of 6–12 months [54,55]. Most of these studies have shown improvement in oxidative stress parameters [49,50,51,52,53,55] but no changes in EDSS [50,51,53,54,55] and new lesions detected by MRI [54]. Some have shown improvement in sleep quality [50] or fatigue scores [52]. The sample size of these studies is low.

Clinical trials with EGCG in MS patients have shown liver tolerance problems [61,62]. Despite one pilot study combining EGCG and coconut oil showing improvement in gait and balance [64] and other improvements in several metabolic parameters [65], two double-blind placebo-controlled studies involving an important number of patients and with sufficiently prolonged follow-up have shown a lack of improvement in clinical and radiological parameters with EGCG [61,63].

The only published clinical trial on the effect of curcumin as add-on therapy in MS patients under IFN β-1a showed a lack of clinical and radiological efficacy [73]. Therapeutic attempts with pentoxifylline, all of which have been shown in pilot studies, have shown a lack of clinical efficacy and, in general, poor tolerance [90,91,92,93,94,95]. Double-blind, placebo-controlled clinical trials in a short series of patients on MS have shown a lack of efficacy in improving clinical parameters with fish oil [110,111] and pomegranate seed oil [113], although there are discrepancies about the improvement or not in markers of inflammation [110,111].

Short-term clinical trials with CoQ_10_ or its derivative idebenone have also shown a lack of clinical efficacy [116,117,118,119,120], except as an anecdotal fact, a slight improvement in fatigue and depression in some patients in a single study [118], although some described improvement in markers of oxidative stress [116] or inflammation [117]. Some randomized double-blind, placebo-controlled clinical trials have shown slight improvement in some clinical [129,132,133] or biochemical parameters [134] after long-term treatments with a mixture of vitamins A, E, γ-tocopherol associated with other antioxidants [137] or vitamin A [132,133,134], although without improvement on global scales such as EDSS, in relapse rate, or radiological lesions [132].

Promising results in some clinical or biochemical parameters found in preliminary studies with *N*-acetyl-cysteine [162,163,164] need to be confirmed in a long-term randomized, placebo-controlled study currently underway [165].

Cytoflavin [177], L-carnosine [187], carnitine [188], and other antioxidant substances summarized in Table 2 [280,281,282,283,284,285,286,287,288,289,290,291,292] have shown improvement in some clinical or biochemical parameters, but only in preliminary studies. Despite having demonstrated positive effects in experimental models, to date, no studies have been published on the possible effects in patients with MS of resveratrol, bilirubin, NOS inhibitors and NO scavengers, PPAR-gamma agonists, edaravone, phycobiliproteins, antidiabetic drugs, caffeic acid, and histone deacetylase inhibitors.

According to the previously presented evidence, it is likely that the mechanism of action of many of the antioxidant substances tested in experimental models of MS is not exclusively related to their antioxidant action. In fact, many of them have shown indirect (reducing pro-inflammatory substances) and/or direct (increasing anti-inflammatory substances) anti-inflammatory actions, as well as the ability to decrease demyelination and reduce axonal damage. The possible interactions between these mechanisms are represented in Figure 16.

In conclusion, many diverse antioxidant substances have shown beneficial effects on experimental models of MS. However, the results of studies with antioxidant therapies in patients with MS have been to date inconclusive. Future long-term, prospective, multicenter, randomized, placebo-controlled trials involving an important number of patients with MS, testing the possible efficacy of several of the antioxidants that have shown beneficial effects in experimental models of MS would be required to clarify the possible value of such treatments trying to improve symptoms and slow the progression of this disease.

## Figures and Tables

**Figure 1 biomolecules-14-01266-f001:**
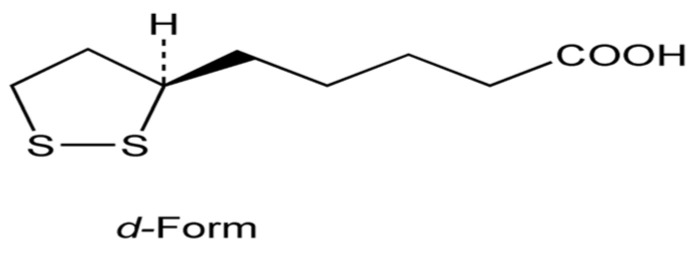
Chemical structure of alpha-lipoic acid (ALA).

**Figure 2 biomolecules-14-01266-f002:**
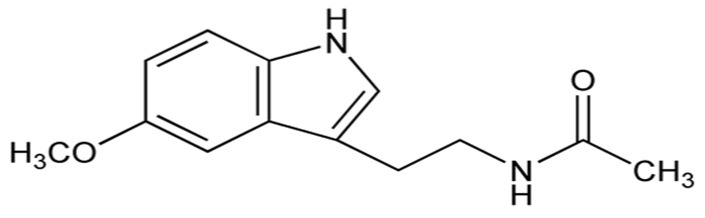
Chemical structure of melatonin.

**Figure 3 biomolecules-14-01266-f003:**
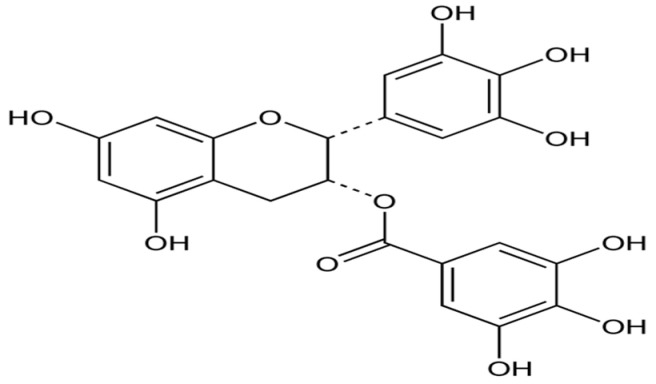
Chemical structure of Epigallocatechin-3-gallate (EGCG).

**Figure 4 biomolecules-14-01266-f004:**
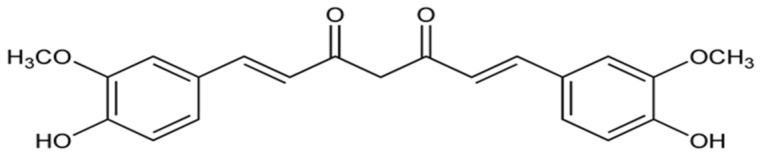
Chemical structure of curcumin.

**Figure 5 biomolecules-14-01266-f005:**
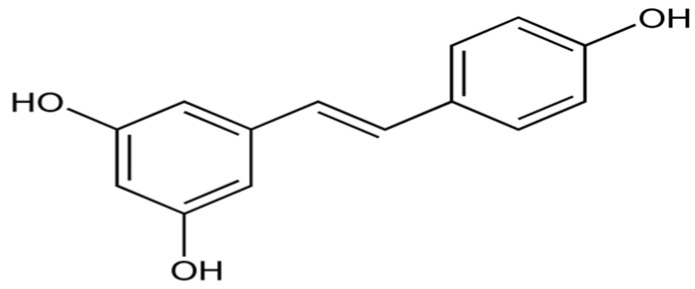
Chemical structure of resveratrol.

**Figure 6 biomolecules-14-01266-f006:**
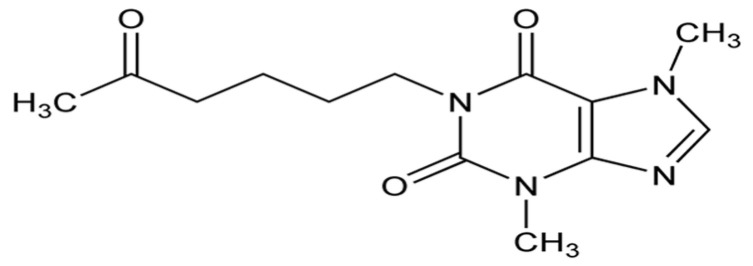
Chemical structure of Pentoxifylline.

**Figure 7 biomolecules-14-01266-f007:**
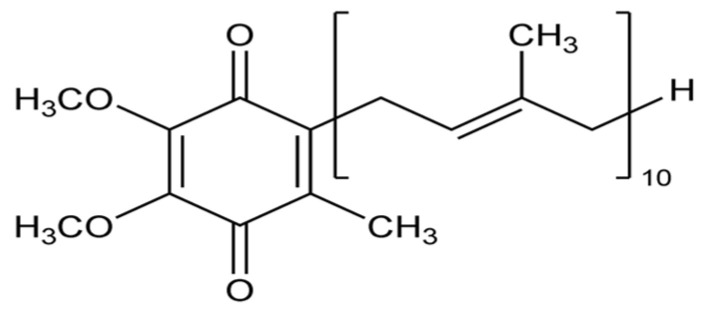
Chemical structure of Coenzyme Q_10_ (CoQ_10_).

**Figure 8 biomolecules-14-01266-f008:**
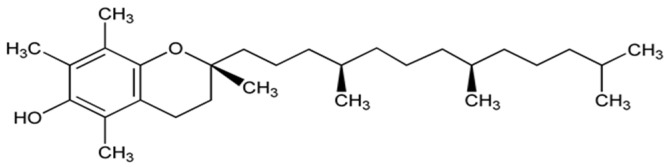
Chemical structure of alpha-tocopherol.

**Figure 9 biomolecules-14-01266-f009:**
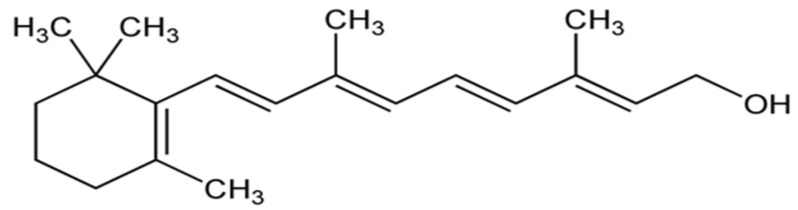
Chemical structure of vitamin A (retinol).

**Figure 10 biomolecules-14-01266-f010:**
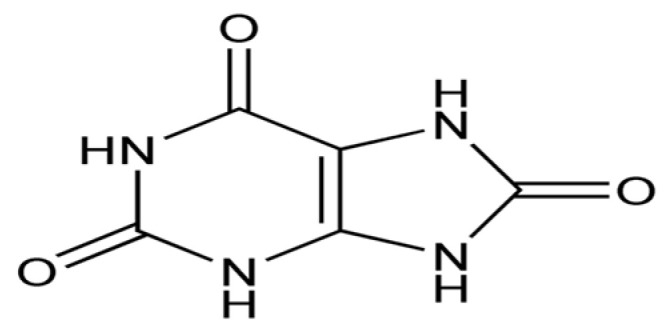
Chemical structure of uric acid.

**Figure 11 biomolecules-14-01266-f011:**
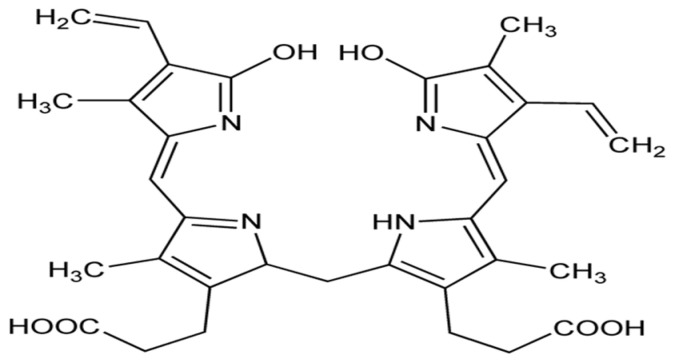
Chemical structure of bilirubin.

**Figure 12 biomolecules-14-01266-f012:**
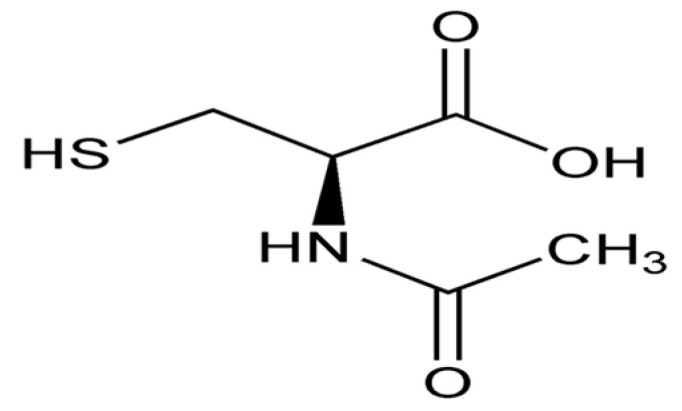
Chemical structure of N-acetyl-cysteine.

**Figure 13 biomolecules-14-01266-f013:**
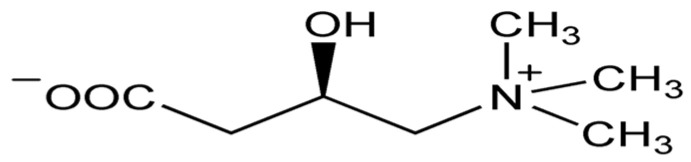
Chemical structure of carnitine (3-hydroxy-4-trimethylaminobutyrate, L-carnitine, levocarnitine).

**Figure 14 biomolecules-14-01266-f014:**
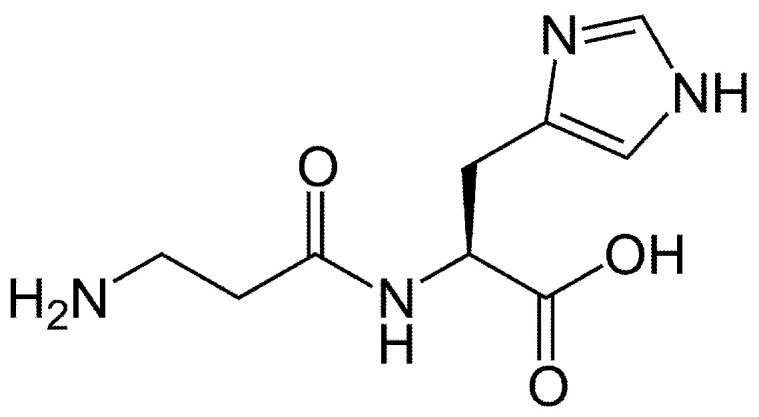
Chemical structure of carnosine (beta-alanyl-L-histidine).

**Figure 15 biomolecules-14-01266-f015:**
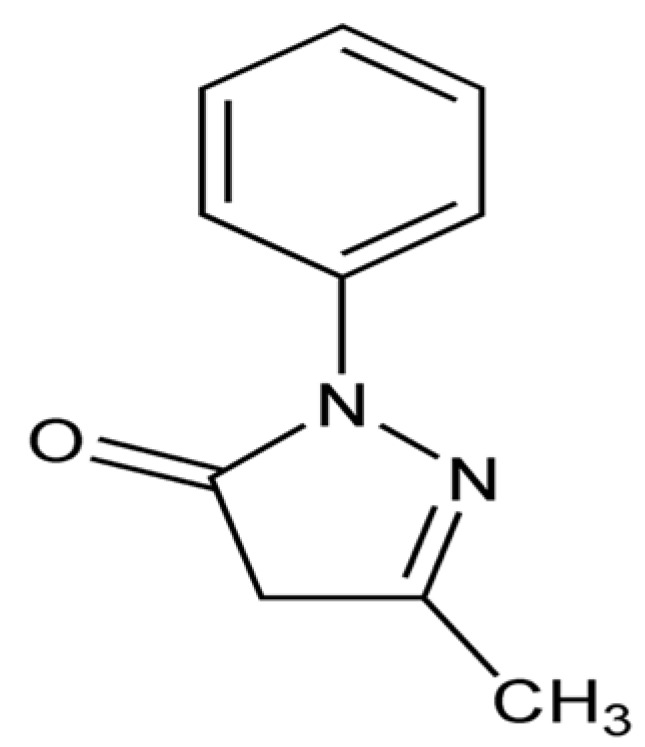
Chemical structure of edaravone.

**Figure 16 biomolecules-14-01266-f016:**
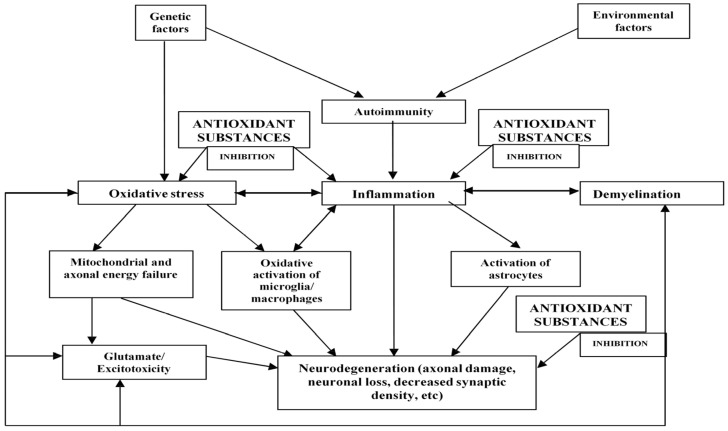
Proposed mechanisms of action of antioxidant substances on the pathogenetic mechanisms of multiple sclerosis (modified from [12]).

**Table 1 biomolecules-14-01266-t001:** Other antioxidant compounds tested in experimental models of MS.

Drug Class	Author, Year [Ref]	Animal Model	Main Findings
Combined matrix metalloproteinase (MMP) and TNF-alpha inhibitor	Clements et al., 1997 [203]	Male Lewis rats aged 5–8 weeks with EAE	Reduction of clinical signs and weight loss, a decrease in MMP activity in the CSF, a marked increase of the MMP matrilysin, and a modest increase in the MMP 92 kDa in the spinal cord.
Matrix metalloproteinase (MMP) inhibitor	Liedtke et al., 1998 [204]	SJL/J mice with EAE	Clinical improvement (blocking and reversal of acute disease, reduced number of relapses, improvement in clinical scores). Significant decrease in demyelination and glial scarring and in central nervous system gene expression for *TNF alpha* and *fasL* and an increase in *IL-4* expression.
EUK-8 (synthetic catalytic scavenger of oxygen-reactive metabolites)	Malfroy et al., 1997 [205]	PL/J (H-2u) mice aged 3–8 months with EAE	Complete recovery after 40 days of EAE mice pretreated with EUK-8 and significant improvement after treatment with EUK-8 4 days after EAE induction.
Drugs acting on heme oxygenase 1 (HO-1)	Liu et al., 2001 [206]	Adult male Lewis rats with EAE	Hemin (HO-1 inducer) improved the clinical course of EAE and reduced inflammatory changes in the spinal cord, while tin mesoporphyrin (HO-1 inhibitor) worsened the clinical course of EAE and increased non-significantly inflammatory changes in the spinal cord.
t-butyl hydroxy anisole (phase 2 enzyme inducer)	Mohamed et al., 2002 [207]	Lewis rats aged 10–12 weeks with EAE	Clinical improvement of EAE, reduction in perivascular lymphocyte infiltration, and increase in red cell GSH concentrations.
Thymoquinone	Mohamed et al., 2002 [208]	Female Lewis rats with EAE	Prevention (if administered previously to EAE induction) or reversal (if administered after EAE induction) of clinical symptoms, perivascular inflammation, and decrease in spinal cord GSH levels.
Silibinin (*Silybum marianum* fruit extract)	Min et al., 2007 [209]	Female, 6-week-old C57BL/6 mice	Prevention or attenuation of clinical signs. Reduction of histological signs of demyelination and inflammation in the spinal cord. Down-regulation of the inflammatory cytokines IL-12 and IL-2 and slight upregulation of the anti-inflammatory cytokine IL-4 in a concentration-dependent manner in splenocytes.
ABS-75 (fullerene derivative)	Basso et al., 2008 [210]	Non-obese diabetic (NOD) mice with EAE	Clinical improvement associated with reduced axonal loss and demyelination, and halted oxidative injury, CD11b+ infiltration, and CCL2 expression in the spinal cord.
GEMSP (mixture of fatty acids, vitamins, and amino acids or their derivatives; and Poly-L-Lysine)	Mangas et al., 2008 [211]	10–11 weeks Lewis 1A female rats with EAE	Complete clinical reversal of EAE, reduction in brain leukocyte infiltration, and preserved immunoreactivity to conjugated methionine antibodies in the motoneurons of the ventral horn.
Genistein (derivative of soy)	De Paula et al., 2008 [212]	8–12 weeks old female C57BL/6 mice with EAE	Significant improvement in clinical symptoms, modulating pro- and anti-inflammatory cytokines in the CNS and splenocytes (upregulation of IL-10 and downregulation of IFN-gamma, TNF-alpha, and IL-12), and decreasing rolling and adhering of leukocytes to the CNS.
Tempamine (TMN) encapsulated in the intraliposomal aqueous phase of pegylated nanoliposomes	Kizelsztein et al., 2009 [213]	Female C57BL/6 mice with EAE	Significant improvement of clinical symptoms and reduction in the expression of mRNA for IFN-gamma and TNF-alpha in the brain.
Ethanol extract of saffron (*Crocus sativus* L.)	Ghazavi et al., 2009 [214]	8-week-old male C57BL/6 mice with EAE	Significant clinical improvement, decrease of inflammatory changes in the brain, increase in serum TAC, and non-significant changes in serum NO levels.
Ethylene diamine tetra acetic acid (EDTA)	Mosayebi et al., 2010 [215]	6–8-week-old male C57BL/6 mice with EAE	Significant delay in the time of onset and a significant reduction in severity of the EAE. Reduction in inflammatory changes and density of mononuclear infiltration in the CNS. Increase in total serum antioxidant power, decrease in serum NO levels and decrease in levels of IFN-gamma in cellular cultures of splenocytes.
Aloe vera	Mirshafiey et al., 2010 [216]	6–8-week-old male C57BL/6 mice with EAE	Significant delay in the time of onset and a significant reduction in severity of the EAE. Reduction in inflammatory changes and density of mononuclear infiltration in the CNS. Increase in total serum antioxidant power, decrease in serum NO levels and decrease in levels of IFN-gamma in cellular cultures of splenocytes.
Erythropoietin	Chen et al.,, 2010 [217]	6–8-week-old male C57BL/6 mice with EAE	Significant clinical improvement. Significant increase in expression of HO-1 mRNA in the spleen and the CNS. Significantly decrease in the expression of interferon-γ, interleukin (IL)-23, IL-6, and IL-17 mRNA, and increase in the expression of IL-4 and IL-10 mRNA in CNS. Significant decrease in the ratio of both T helper type 1 (Th1) and Th17 lymphocyte subsets isolated from CNS and an increase in the ratio of splenic regulatory CD4 T cells.
Hydralazine (acrolein scavenger)	Leung et al., 2011 [218]	8-week-old male C57BL/6 mice with EAE	Significant delay and improvement of clinical symptoms. Significant decrease of demyelination in the spinal cord.
Spermidine	Guo et al., 2011 [219]	6–8-week-old female C57BL/6 mice with EAE	Significant improvement of clinical symptoms, including optic neuritis. Significant improvement of demyelination in the spinal cord and optic nerve and prevention of cell loss and apoptosis in the retinal ganglion cell layer.
L-cycloserine (inhibitor of Ceramide synthase 6, CS6)	Schiffmann et al., 2012 [220]	8–10 week-old female C57BL/6 or IFN-γ KO with EAE	Remission of the disease related to prevention of the increase in C(16:0)-Cer (reflecting a decrease in CS6 expression) and iNOS/TNF-α expression.
Aqueous extract of North American ginseng	Bowie et al., 2012 [221]	C57BL/6 6 week-olf female mice with EAE	Significant decrease in clinical signs of EAE, levels of circulating TNF-α, and CNS immunoreactive iNOS and demyelination scores.
Safinamide and flecainide (sodium channel-blocking agents)	Morsalli et al., 2013 [222]	Male and female Dark Agouti rats	Pretreatment and treatment with these drugs caused significant improvement of neurological deficits and protection against neurological deficit and axonal degeneration associated with reduction in the CNS of the activation of microglia/macrophages.
MitoQ (mitochondria-targeted antioxidant)	Mao et al., 2013 [223]	10-week-old C57BL/6 mice with EAE	Pretreatment and treatment with this drug caused significant improvement of neurological deficits, reversed the increase of mRNA expression of CD4, CD8, CD11b, IL1, IL6, IL10, IL17, TNFα, NOS2, and NFkB in the spinal cord, and induced upregulation of the anti-apoptotic gene *STAT3* in the spinal cord.
Sulforaphane	Li et al., 2013 [224]	8–10 week-old female C57BL/6 mice with EAE	Inhibition of development and severity of EAE. Decrease of inflammation and demyelination in the spinal cord. Prevention or reversion of the increased MDA levels expression of MMP-9 and decreased expression of Nrf2, HO-1, and NQO1 in the brain caused by EAE. Reversion of increased IL-17A-secreting Th17 cells in splenocytes.
Astragaloside IV (a natural saponin molecule)	He et al., 2013 [225]	6-week-old female C57BL/6 mice with EAE	Improvement of clinical signs of EAE. Inhibition or reversion of the increase of ROS and pro-inflammatory cytokine levels, downregulation of SOD and GSH-Px activities, increase of iNOS, p53, and phosphorylated tau, and decrease of ratio of Bcl-2/Bax in CNS caused by EAE
Β-asarone modified astragaloside IV (a natural saponin molecule)	Zhao et al., 2023 [226]	6–8 week-old female C57BL/6 mice	Improvement of clinical signs of EAE. Suppression of inflammatory infiltration and astrocyte/microglial activation and reduction of demyelination in the brain.
R-flurbiprofen	Schmitz et al., 2014 [227]	10–12 week old female C57BL6/J mice with primary progressive EAE and 10–12 week female SLJ mice with relapsing-remitting EAE in SJL mice	Prevention of attenuation of clinical signs in both models of EAE. Decrease of immune cell infiltration and microglia activation and inflammation in the spinal cord, brain, and optic nerve and attenuation of myelin destruction and EAE-evoked hyperalgesia. Increase of CD4(+)CD25(+)FoxP3(+) regulatory T cells, CTLA4(+) inhibitory T cells, and IL-10, Important decrease of upregulation of pro-inflammatory genes in the spinal cord. Association of these effects to the increase of plasma and cortical endocannabinoids and decreased spinal prostaglandins.
Apocynin (NADPH oxidase assembly inhibitor)	Choi et al., 2015 [228]	Female C57BL/6 mice	Significantly marked clinical improvement of EAE associated with suppression of demyelination, reduced infiltration by CD4, CD8, CD20, and F4/80-positive cells, reduced induction of pro-inflammatory cytokines in cultured microglia, and inhibition of free radicals production and blood–brain barrier disruption.
Flavocoxid (dual cyclooxygenase and 5-lipoxygenase inhibitor)	Kong et al. 2016 [229]	Female C57BL/6 mice 6–10 weeks old	Significant clinical improvement of EAE. Decrease in COX2, 5-LO, IL12, IL23, IL17, IFNγ and increase in IL10 in the spinal cord. Decrease of MHCII, CD40, CD80, CD86, COX2, 5-LO, IL12 and IL6, and increase of IL-10 in primary macrophages. Increase of Arg1, Ym1, CD206, TGFβ1, IL10 and decrease of iNOS and IL-12 in macrophages and microglia. Inhibition of Th1/Th17 differentiation.
Hydrogen-rich water	Zhao et al., 2016 [230]	C57BL/6 mice 6–8 weeks old	Significant clinical improvement of EAE (delay and reduction of clinical severity). Inhibition of inflammatory infiltration and demyelination of CNS. Prevention of infiltration by CD4^+^ population and inhibition of Th17 cell differentiation in CNS and peripheral immune organs.
Arctigenin (lignan present in the extract from *Arctium lappa*)	Li et al., 2016 [231]	Female C57BL/6 mice 6–8 weeks old	Significant clinical improvement of EAE. Decreased inflammation and demyelination, inhibition of Th1 and Th17 cells in peripheral immune organs. Suppression of the Th1 cytokine IFN-γ, the transcription factor T-bet, the Th17 cytokines IL-17A, IL-17F, and the transcription factor ROR-γt. Inhibition of Th17 cells (but not Th1 cells) in the CNS. Activation of s AMPK, inhibition of phosphorylated p38, and upregulation of PPAR-γ.
3H-1,2-dithiole-3-thione (D3T, a potent inducer of antioxidant genes and glutathione biosynthesis)	Kuo et al., 2016 [232]	C57BL/6 mice and Nrf2 deficient mice (*Nrf2^−^*^/*−*^) with its corresponding controls (*Nrf2^+^*^/*−*^) of 7–10 weeks of age	Significant clinical improvement of EAE. Suppression of the activation of dendritic cells. Suppression of IL-23 and co-stimulatory molecules in dendritic cells through induction of Nrf2. Inhibition of Th1 and Th17 differentiation and activation of microglia.
Pramipexole (D2/D3 receptor agonist with antioxidant actions)	Lieberknecht et al., 2017 [233]	Female C57BL/6 mice with EAE	Prevention of development of clinical signs of EAE. Blocking of neuroinflammatory response, demyelination, and astroglial activation in spinal cord. Inhibition of the production of inflammatory cytokines (IL-17, IL-1β, and TNF-α) in peripheral lymphoid tissue. Reversion of effects of EAE in reactive oxygen species, glutathione peroxidase, parkin, and α-synuclein in the spinal cord and striatum.
Despramipexole	Buonvicino et al., 2020 [234]	Female non-obese diabetic NOD/ShiLtj and C57BL/6 mice with EAE	Delay in progression of clinical symptoms. Increase in survival, counteracted reduction of spinal cord mitochondrial DNA content, and reduced spinal cord axonal loss of mice.
TEMPOL (4-hydroxy-2,2,6,6-tetramethyl-piperidine-1-oxyl, membrane-permeable radical scavenger)	Neil et al., 2017 [235]	Female C57BL/6J (Jackson Labs) or SJL/J of 8–10 weeks of age	Prevention of EAE by delayed onset, reduced incidence, and reduction of clinical severity of the disease. Decrease in microglial activation and fewer inflammatory infiltrates and axonal damage. Decrease in production of INF-γ and TNF-α by T-cells, decrease in levels of MHC class II expression and CD40 in myeloid and myeloid-dendritic cells. Increase in CD8+ T cell populations and CD4+FoxP3+ regulatory populations.
β-Caryophyllene (sesquiterpene derivative)	Fontes et al., 2017 [236]	Female C57BL/6 mice 8–12 weeks old	Significant improvement of clinical score and severity of EAE. Significant decrease in the number of inflammatory infiltrates and attenuation of neurological damage. Inhibition of H_2_O_2_, NO, TNF-α, IFN-γ, and IL-17 production.
BJ-2266 (6-thioureido-derivative)	You et al., 2017 [237]	C57BL/6 mice and OT-II mice	Significant suppression of EAE disease progression. Reduction of generation of Th1 and Th17 cells through inhibition of JAK/STAT phosphorylation.
D-aspartate	Afraei et al., 2017 [238]	Female C57BL/6 mice 8 weeks old	Significant attenuation in the severity and delay in the onset of the disease. Reduction of demyelination, inflammation, and degeneration in the CNS. Significant decrease in serum IL-6 levels. Non-significant increase in SOD and glutathione-reductase activities and TAC in serum.
18β-glycyrrhetinic acid	Kamisli et al., 2018 [239]	Female C57BL/6 mice	Significant clinical improvement. Reversion of histological and immunological alterations caused by EAE. Increased brain levels of TBARS levels and decreased GPx, SOD, CAT, and GSH levels in brain tissue. Decreased caspase-3 and IL-17 activity in brain tissue.
Ethanol extract of Glycyrrhizae Radix	Yang et al., 2019 [240]	Female C57BL/6 mice aged 5 weeks	Clinical improvement of EAE signs associated with a decrease of antigen-specific TH1 and TC1 populations and an increase of CD8+IL-17A+ (TC17) population. Reduction in IFN-γ+ T cell populations, the expressions of cell adhesion molecules (CAMs), and secretions of cytokines containing IFN-γ and a chemokine IFN-γ-induced protein 10 (IP-10) in splenocytes in culture. Decrease in NO production and secretion of TNF-α and IP-10 in IFN-γ-stimulated BV2 cells in culture.
Oligomeric proanthocyanidin (OPC), the most effective component of grape seed extract	Wang et al., 2019 [241]	Male C57BL/6J mice with cuprizone-induced demyelination	Attenuation of abnormal behavior reduced demyelination and increased expression of myelin basic protein and expression of O4+ oligodendrocytes in the corpus callosum. Reduction in numbers of B and T cells, activation of microglia in the corpus callosum, and inhibition of secretion of inflammatory factors.
Gold nanoparticles and polyethylene glycol	Aghaie et al., 2019 [242]	Female C57BL/6 mice aged 8–10 weeks with EAE	Significant decrease in the severity of EAE symptoms. Significant decrease in the number and severity of cells’ infiltration and demyelinated lesions in the spinal cord. Significant increase in IL-27 and decrease in IL-23 levels (only for gold nanoparticles).
*Oenothera biennis* L. and *Hypericum perforatum* L. extracts	Selek et al., 2019 [243]	8–10-week-old C57BL/6J mice	Significant decrease in the severity of EAE symptoms. Significant decrease in brain tissue MOG and MBP. Significant decrease in the TOS and OSI values and increase in TAS in brain tissue. Significant decrease in myelin loss and amyloid deposition on vascular walls, in the cytoplasm of the neurons, and in the intercellular space.
Mitochondria-Targeted Antioxidant SkQ1	Fetisova et al., 2019 [244]	In vitro MS model of the primary oligodendrocyte culture of the cerebellum, challenged with lipopolysaccharide (LPS).	SkQ1 accumulated in the mitochondria of oligodendrocytes and microglial cells and prevented LPS-induced inhibition of myelin production in oligodendrocytes.
Ellagic acid (a polyphenol found in numerous fruits and vegetables with antioxidant actions)	Khodaei et al., 2019 [245,246]	6–9 week-old male C57BL/6 mice with cuprizone-induced demyelination	Significant improvement of EAE symptoms. Decrease in MDA levels, increase in TAC, and GSH/GSSG ratio in muscle tissue. Increase in mitochondrial dehydrogenase. SOD and catalase activities, mitochondrial ATP levels and mitochondrial GSH/GSSG ratio, and decrease in mitochondrial depolarization in muscle tissue. Increase in mitochondrial complex I, II, and IV activities in Sirt3 level and *Sirt3* expression in muscle tissue.
Tetrapeptide N-acetyl-ser-asp-lys-pro (Ac-SDKP)	Pegman et al., 2020 [247]	10-week-old female C57BL/6 mice with EAE	Significant improvement of EAE symptoms. Decrease in inflammatory infiltration and demyelination in hippocampus oligodendrocytes. Reduction of reactive oxygen species, MDA levels, and NOS production. Increase in GSH levels, GPx and HO-1, and total antioxidant activities. Decrease in caspase-12 and CHOP expression in the hippocampus-resident oligodendrocytes. Decrease in the activation of the proinflammatory cytokines IL-6 and IL-1β and increase in levels of the anti-apoptotic proteinBcl2.
Acteoside (AC), an active compound from the medicinal herb *Radix rehmanniae*	Li et al., 2020 [248]	10–14-week-olf female C57BL/6 N mice with EAE	Significant improvement and delay of EAE symptoms. Inhibition of inflammation/demyelination, and peripheral activation and CNS infiltration of encephalitogenic CD4+ T cells and CD11b+ activated microglia/macrophages in the spinal cord Decrease in peroxynitrite production down-regulation of the expression of iNOS and NADPH oxidases, and inhibition of neuronal apoptotic cell death and mitochondrial damage in the spinal cords.
GYY4137 (Hydrogen Sulfide Donor)	Lazarević et al., 2020 [249]	Dark Agouti rats or C57BL/6 mice with EAE	Increase in TGF-β expression and production in dendritic cells and significant reduction of IFN-γ and IL-17 production in the lymph node and spinal cord T cells obtained from mice immunized with CNS antigens. Decrease in the proportion of FoxP3+ regulatory CD4+ T cells in the lymph node cells and the percentage of IL-17+ CD4+ T cells in the spinal cord cells after culturing with GYY4137.
Shikonin (extract from plants of the Boraginaceae family)	Nasrollahzadeh Sabet et al. 2020 [250]	Female C57BL/6 mice aged 10–12 weeks with EAE	Significant improvement and delay of EAE symptoms. Significant decrease in the extent of demyelination. Significant reduction in the expression levels of TNF-α, IFN-γ, and Bax, and increase of levels of TGF-β and Bcl2 and the activity of Gpx1.
Ursolic acid (pentacyclic triterpenoid compound)	Yamamoto et al., 2020 [251]	Male C57BL/6J mice with cuprizone-induced demyelination	Significant improvement in motor dysfunction and demyelination. Increase of IGF-1 levels in the demyelinating lesions.
Herbal extract of *Melilotus officinalis*	Hassani et al., 2020 [252]	Female C57BL/6 mice with EAE	Prevention or attenuation of the clinical signs of the disease. Decreased demyelination in the corpus callosum. Significant decrease in the gene expression of pro-inflammatory cytokines (*IL-6*, *TNF-α*, and *IFN-γ*). Increase in the gene expression of *GPx* and *CAT*.
Piperine, a main bioactive alkaloid of black pepper	Nasrnezhad et al., 2021 [253]	Female Lewis rats with EAE	Significant improvement in neurological deficits and progression. Significant decrease in the extent of demyelination, inflammation, immune cell infiltration, microglia, and astrocyte activation in the spinal cord. Significant decrease in the gene of pro-inflammatory cytokines (*TNF-α*, *IL-1β*) and *iNOS* and enhanced *IL-10*, *Nrf2*, *HO-1*, and *MBP* expressions. Increase in TAC and reduction of MDA levels in the CNS. Reduction of caspase-3 (apoptosis marker) and enhancement of brain-derived neurotrophic factor (BDNF) and NeuN-expressing cells in the CNS.
Diosgenin (herbal from Dioscora species)	Zeinali et al., 2021 [254]	Female C57BL/6 mice aged 10–12 weeks with EAE	Significant clinical improvement. Decrease of inflammation, demyelination, and axonal degeneration severity in lumbar spinal cord in EAE. Lack of effect on modification of neuroprotective factors decreased by EAE. Decrease in serum and tissue levels of TNFα and MMP-9 with no effect on increased level of IL-17 in EAE.
Withametelin (phytosterol derivative)	Khan et al., 2021 [255]	Female Swiss mice of 8–12 weeks of age with EAE	Significant clinical improvement. Reversion of histopathological alteration of the brain, spinal cord, eye, and optic nerve. Reduction of H_2_O_2_-induced cytotoxicity and oxidative stress in a dose-dependent manner. Increase in the expression level of nuclear factor-erythroid-related factor-2 (Nrf2), heme-oxygenase-1 (HO-1), and reduction of the expression level of the Kelch-like-ECH-associated-protein-1 (keap-1), inducible-nitric-oxide-synthase (iNOS) in the CNS. Decrease in the expression level of toll-like-receptor 4 (TLR4), nuclear-factor-kappa-B (NF-κB), and increase in the expression level of IkB-α in the CNS.
Memantine (uncompetitive NMDA receptor antagonist)	Dąbrowska-Bouta et al. 2021 [256]	Female Lewis rats with EAE	Significant clinical improvement. Significant decrease in MDA levels and SOD1 and SOD2 expression, and increase in –SH groups levels in brain tissue.
Tibolone (synthetic steroid whose metabolites have potent antioxidant action)	Mancino et al., 2022 [257]	Female C57BL/6 mice with EAE	Significant clinical improvement. Reversion of the astrocytic and microglial reaction and reduction of the hyperexpression of TLR4, HMGB1, and NLR family pyrin domain containing 3-caspase 1-interleukin-1β inflammasome activation in the spinal cord. Attenuation of the Akt/nuclear factor kappa B pathway and decrease in the white matter demyelination area in the spinal cord.
Herbal plants *Nepeta hindustana* L., *Vitex negundo* L., and *Argemone albiflora* L.	Rasool et al., 2022 [258]	Wistar rats with lipopolysaccharide-induced MS	Neuroprotective effect regarding remyelination of axonal terminals and oligodendrocytes migration reduced lymphocytic infiltrations and reduced necrosis of Purkinje cells in histopathological examination. Reverse of changes in serum levels of inflammatory markers (MMP-6, TNF-α, AOPPs, AGEs, NO, IL-17 and IL-2), MDA, antioxidant (SOD, GSH, CAT, GPx), DNA damage markers (Isop-2α, 8OHdG), RAGE, caspase-8, and p38, induced by lipopolysaccharides.
Acetyl-11-keto-β-boswellic acid (a major component of *Boswellia serrata*)	Nadeem et al., 2022 [259]	Female SJL/J mice 8–9 weeks old with EAE	Significant clinical improvement. Downregulation of inflammatory markers in CD11c + DCs (p-NF-κB, iNOS, and nitrotyrosine) and CNS (p-NF-κB, iNOS, nitrotyrosine, lipid peroxides, and total antioxidant capacity).
Acetyl-11-keto-β-boswellic acid (a major component of *Boswellia serrata*)	Upadhayay et al., 2022 [260]	Wistar rats with EAE	Significant improvement in clinical parameters improvement and reversion of the EAE-induced histopathological changes in the brain. Reversal of the decreased Nrf2 and HO-1 levels in the CSF and brain homogenate caused by EAE-induction. Decrease in caspase-3 and Bax, and increased Bcl-2 Levels in brain homogenates. Reversion of decrease in acetylcholine, dopamine, and serotonin, and increase in glutamate in brain homogenates induced by EAE. Reversion of the increase in AchE, MDA, and nitrite levels and reduction of GSH, SOD, and catalase levels induced by EAE. Decrease in TNF-αand IL-1β levels in brain homogenates.
*Urtica dioica* extract	Namazi et al., 2022 [261]	Male C57BL/6J mice 8 weeks old with cuprizone-induced demyelination	Significant decrease of demyelination with high doses. Significant decrease of MDA and HSP-70 brain levels with no effect on HSP-60.
Nebivolol	Naeem et al., 2022 [262]	Male C57BL/6J mice with cuprizone-induced demyelination	Significant clinical improvement and reversion of the EAE-induced histopathological changes. Modulation of microglial activation status by suppressing M1 markers (Iba-1, CD86, iNOS, NO, and TNF-α) and increasing M2 markers (Arg-1 and IL-10). Inhibition of NLRP3/caspase-1/IL-18 inflammatory cascade. Decrease of MDA levels and increase of CAT and SOD activities.
IM253 (herbal compound)	Esmaeilzadeh et al., 2022 [263]	Female C57Bl/6 mice 10–12 weeks old with EAE	Significant clinical improvement and delay of symptoms of EAE. Significant decrease in inflammation and demyelination. Significant reduction in the expression of pro-inflammatory cytokine coding genes (*IL-6*, *TNF-α*, *IFN-γ*, and *IL-17*) and increase in the expression of anti-inflammatory and anti-oxidant enzyme coding genes (*TGF-β*, *GPX-1*, and *CAT*) and TAC.
*Zingiber officinale* (ginger) extract	Moradi et al., 2022 [264]	Male Wistar rats 8-week old with cuprizone-induced demyelination	Significant clinical improvement and decrease of demyelination with high doses. Significant expression and increased levels of MBP and OLIG2.
Ginger extract	Kamankesh et al. 2023 [265]	Female C57Bl/6 mice 8 weeks old with EAE	Significant clinical improvement without a decrease in lymphocyte infiltration. Reduction of gene expression levels of the inflammatory cytokines IL-17 and IFN-γ. Significant increase of Treg cells. Significant reduction of serum NO levels.
Grape seed extract	Mabrouk et al., 2022 [266]	Female C57Bl/6 mice 6 weeks old with EAE	Significant clinical improvement and decrease of demyelination induced by EAE. Significant decrease in MDA and carbonyl protein levels, increase in SOD1, SOD2, GPx, and CAT activities, decrease in GFAP expression, and increase in SIRT1 expression in the brain and spinal cord (reversing the effects of EAE).
Grape Seed Extract	Wang et al., 2023 [267]	C57BL/6 mice with EAE	Significant improvement of clinical symptoms of EAE. Inhibition of spinal cord demyelination and inflammatory cell infiltration. Inhibition of the secretion of TNF-α, IL-1 β, IL-6, IL-12, IL-17, and IFN-γ cells and reduction of the differentiation of Th1 and Th17 mediated by CD3 and CD28 factors in the spleen. Inhibition of the infiltration of CD45+CD11b+ and CD45+CD4+ cells, differentiation of Th1 and Th17 (*p* < 0.05), reduction of secretion of inflammatory factors, and prevented the activation of microglia in the CNS.
Daphnetin (7,8-dihydroxy-coumarin)	Soltanmoham-madi et al., 2022 [268]	Female C57Bl/6 mice 8 weeks old with EAE	Significant clinical improvement and decrease of demyelination and lymphocyte infiltration in the CNS. Increase in the expression of anti-inflammatory cytokines and transcription factors (IL-4, IL-10, IL-33, GATA3, TGF-β, FoxP3), and reduction in the production of pro-inflammatory cytokines and transcription factors (IFN-γ, STAT4, T-bet, IL-17, STAT3, ROR-γt, TNF-α) in the CNS. Significant increase of the ratio of spleen Treg cells and the levels of IL-4, IL-10, TGF-β, and IL-33, and reduction of the levels of IFN-γ, TNF-α, and IL-17 in splenocytes.
Gallic (3,4,5-tri hydroxybenzoic) acid	Stegnjaić et al., 2022 [269]	Dark Agouti (DA) rats 5–7 months old with EAE	Significant clinical improvement and decrease of demyelination and lymphocyte infiltration in the CNS. Decrease in the release of IL-17, IFN-γ, and NO, decrease in the proportion of OX40^+^ or CD25^+^ CD4^+^ and Th17 cells in the spinal cord. Inhibition of NO, IL-6, and TNF release from macrophages and microglia.
*Enterococcus durans* (probiotic)	Samani et al., 2022 [270]	Female C57Bl/6 mice 8–10 weeks old with EAE	Significant decrease of demyelination and lymphocyte infiltration in the CNS. Significant decrease of the inflammatory cytokines IL-17, IFN-γ, and MMP-9, and increase in the MBP levels at the spinal cord
Ilepcimide (antiepilepsirine)	Xu et al., 2023 [271]	Female C57Bl/6 mice 8–10 weeks old with EAE	Ameliorates demyelination, blood–brain barrier leakage, and infiltration of CD4+ and CD8+ T cells.
Cannabigerol (phytocannabinoid)	Feinshtein et al., 2023 [272]	Female C57Bl/6 mice 8 weeks old with EAE	Significant clinical improvement. Reversion of enhanced neuronal loss and areas stained for GFAP in the spinal cord.
L-Theanine (antioxidant present in green tea)	Khosravi-Nezhad et al., 2023 [273]	Female C57Bl/6 mice 4–6 weeks old with cuprizone-induced demyelination	Significant clinical improvement. Reversion of the increase of MDA and the decrease in SOD, GPx, and TAC serum levels induced by cuprizone.
Vitamin D	Haindl et al., 2023 [274]	Male Dark Agouti rats aged 10–12 weeks with EAE	Significant reduction in myelin loss, microglial activation, and number of apoptotic cells. Significant reduction of seri, levels of oxidized lipid markers and NfL, and increase in serum TAC and protective polyphenols.
Peroxiredoxin 6	Lunin et al., 2023 [275]	Female SJL/J mice, aged 3 months	Significant clinical improvement. Decrease of EAE-induced *NOX1* and *NOX4* gene expression in brain tissue
Bifidobacterium breve and *Lactobacillus casei* (probiotics)	Hasaniani et al., 2023 [276]	Male Wistar rats aged 5–6 weeks with cuprizone-induced demyelination	Significant clinical improvement. Increase in *HO-1* and *Nrf-2* gene expression, reduction of MDA levels, and increase in TAC in demyelinated corpus callosum.
Sinomenine (isoquinoline alkaloid isolated from *Sinomenii acutum*)	Fan et al., 2023 [277]	C57Bl/6 mice 8 weeks old with EAE	Significant amelioration of EAE severity. Reduction in the demyelination, axonal damage, and inhibition of inflammatory cell infiltration in the spinal cord. Decrease in the inflammatory cytokines TNF-α and IL-1β, MCP-1, and GM-CSF expression in the spinal cord and serum, increase in anti-inflammatory cytokine IL-10 expression in the spinal cord, and suppression of microglia and astrocytes activation in EAE mice. Inhibition of oxidative stress (reduction in MDA and advanced oxidation protein products (AOPP) levels in the spinal cord) via the activation of the Nrf2 signaling pathway (increase in expression of HO-1, NQO-1, and Nrf2 expression in the spinal cord. Suppression of lipopolysaccharide (LPS)-induced microglial activation and the production of pro-inflammatory factors in vitro.
Auranofin (thiol-reactive gold-containing compound)	Al-Kharashi et al., 2023 [278]	Female SJL/J mice aged 9–10 weeks with EAE	Significant amelioration of the clinical features. Increase in thioredoxin reductase (TrxR) activity, decrease in Nrf2 signaling, downregulation of the p-NFkB, decrease of iNOS mRNA expression, IL-6 mRNA levels, lipid peroxides levels, and myeloperoxidase activity in the brain and Peripheral Myeloid Immune Cells.
Vanillic acid	Safwat et al., 2023 [279]	Male Sprague–Dawley 12–16 weeks with cuprizone-induced demyelination	Significant improvement in alterations of nerve conduction velocity and demyelination deterioration of the sciatic nerve fibers induced by cuprizone. Significant improvement of the increase in IL1-7 level, expression of INF-γ and caspase-3, and the reduction in the expression of MBP, Nrf2, and HO-1 induced by cuprizone in the sciatic nerve.
Docosahexaenoic acid	Muñoz-Jurado et al., 2024 [280]	Male Dark Agouti rats with EAE	Significant clinical improvement. Decrease in lipid peroxides, carbonylated proteins, nitric oxide, and GSSG, and reduction of GSH levels in brain, spinal cord, blood, and various tissues.
Nicorandil (ATP-sensitive potassium channels opener) + carvedilol (α and β adrenoceptor antagonist)	Mustafa et al., 2024 [281]	Male Sprague–Dawley rats with EAE	Significant clinical improvement. Significant improvement of demyelination. Downregulation of TLR4/MYD88/TRAF6 signaling cascade with inhibition of (pT183/Y185)-JNK/p38 (pT180/Y182)-MAPK axis with reduction of IL-1β, IL-6NF-κB, and TNF-α in the brain. Increase of SOD activity and Nrf2 content and reduction of MDA content of the brain. Inhibition of gene expression for *CD4*, *IL-17*, *IL-23*, and *TGF-β*. Anti-apoptotic effect by decreasing Bax protein expression and caspase-3 content and increasing Bcl-2.
Purmorphamine (Smo-Shh/Gli activator)	Prajapati et al., 2024 [282]	Wistar rats of both sexes with EAE	Significant improvement of clinical parameters and histological changes Significant decrease in the levels of Smo-Shh in brain homogenate, blood plasma, and cerebrospinal fluid. Decrease of brain and plasma TNF-α and IL-1β levels. Decrease of MDA, nitrite, and AchE and increase of reduced glutathione, SOD, and CAT brain levels. Reversion of the increase in the apoptosis-related markers caspase-3 and Bax and the decrease of Bcl-2 anti-apoptotic protein induced by EAE in the brain and plasma. Reversion of the increased neurofilament protein levels in the brain. Increased brain levels of acetylcholine, dopamine, and serotonin and decrease of glutamate.
Arbutin (glycosylated derivative of hydroquinone present in the leaves and bark of bearberry and other plants)	Ashrafpour et al., 2024 [283]	Adult male Wistar rats with demyelination induced by injection of lysolecithin into the hippocampus	Significant improvement of memory impairment. Reduction of demyelination and astrocyte activation, and increase of MBP. Suppression of pro-inflammatory markers (IL-1β, TNF-α) and increase of anti-inflammatory cytokine IL-10. Decreased iNOS and increase of anti-oxidative factors (Nrf2, HO-1) and BDNF expression.
Cerium oxide nanoparticles	Nguyen et al., 2024 [284]	Female BALB/c mice aged 6–8 weeks and female C57BL/6 mice aged 9–10 weeks with EAE	Significant clinical improvement. Reversion of the increase in expression of T cells, dendritic cells, and resident microglia in the CNS and in IFN-γ expression among CD4+ T cells sampled from spleen tissue induced by EAE. Reduction in demyelination in the spinal cord.

**Table 2 biomolecules-14-01266-t002:** Other antioxidant compounds tested in patients with MS.

Drug	Author, Year [Ref]	Study Design	Main Findings
MS14 (natural herbal-marine drug)	Ahmadi et al., 2010 [285]	Open-label study involving 14 MS patients	Improvement of quality of life of patients.
Vegetable extract containing *Lipia citriadora*	Mauriz et al., 2013 [286]	Randomized prospective placebo-controlled study involving 9 MS patients (5 under low-fat diet and antioxidants, 4 under low-fat diet)	Decrease of serum levels of C reactive protein isoprostane 8-iso-PGF2α and interleukine IL-6 and increase or catalase activity in blood.
*Saccharomyces boulardii* (probiotic)	Aghamohammadi et al., 2019 [287]	4-month, double-blind, randomized controlled two-group parallel trial involving 50 MS	This trial is ongoing. The study assesses changes in mental health evaluated by the 28-item General Health Questionnaire. Secondary endpoints include changes in life quality, fatigue, pain, serum markers of inflammatory stress, and oxidative stress.
Ozone therapy	Izadi et al., 2020 [288]	Open-label study involving 20 MS patients	Significant decrease in the frequency of Th17 cells, the expression levels of RORγt and IL-17, miR-141, and miR-155 in post-treatment compared to pre-treatment condition. Significant reduction in the secretion level of IL-17 in treated patients.
CaNa2EDTA and multivitamin complex with redox glutathione	Vezzoli et al., 2023 [289]	Open-label study involving 18 MS patients	Decrease in ROS production, and increase in TAC, SOD, and CAT activity in plasma. Decrease in GSSG and increase in reduced GSH in erythrocytes. Decrease of 8-iso PGF2α, 8-OH-dG, and NO metabolites in urine. Decrease in IL-6, TNF-α, and sICAM plasma levels.
Synbiotics supplement and anti-inflammatory antioxidant-rich diet compared to placebo and their usual diet	Moravejolahkami et al., 2023 [290]	4-month single-center, single-blind, randomized, controlled clinical trial involving 70 MS patients with PPMS, SPMS, or progressive-relapsing MS	Clinical improvement in fatigue, pain, sexual function, and bowel/bladder status.
Ellagic acid (a polyphenol found in numerous fruits and vegetables with antioxidant actions)	Hajiluian et al., 2023 [291]	12-week randomized, triple-blind, placebo-controlled trial involving 50 MS patients with mild to moderate depressive symptoms	Significant decrease in depression score, serum IFN-γ, NO, and cortisol levels, and *indoleamine 2.3-dihydrooxygenase 1* (*IDO*) gene expression. Increase in serum levels of BDNF and serotonin. Lack of changes in serum Nrf2 levels.
Saccharomyces boulardii (probiotic)	Asghari et al., 2023 [292]	4-month prospective randomized double-blinded clinical trial involving 40 RRMS patients	Significant improvement in scales measuring pain, fatigue, quality of life, and somatic and social dysfunction on a General Health Questionnaire. Significant decrease in plasma levels, high-sensitivity C-reactive protein (hs-CRP), and increase of TAC. Non-significant decrease in plasma MDA levels.

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
