# Peer review of "Antioxidant Therapies in the Treatment of Multiple Sclerosis"

_biomolecules, 2024, doi:10.3390/biom14101266_

Round 1

Reviewer 1 Report

Comments and Suggestions for Authors

This is a narrative review with lots of information brought together.

Authors have put in a lot of effort to bring the references and information together for this narrative review. and therefore, this article addresses the width of the topic well.

I note some molecules are extensively reviewed whilst others not, and therefore it may not represent a general review of all relevant molecules but possibly emphasises those molecules that authors have been most interested in. This is the most criticism of the article, and therefore authors may wish to address this.

Comments on the Quality of English Language

Good

Author Response

I note some molecules are extensively reviewed whilst others not, and therefore it may not represent a general review of all relevant molecules but possibly emphasises those molecules that authors have been most interested in. This is the most criticism of the article, and therefore authors may wish to address this.

Thank you for your feedback. We acknowledge your observation regarding the emphasis on certain molecules. The selection of molecules reviewed in detail was based on the current availability of data and their relevance to the topic. The length of the text for all of them is greater or lesser depending on the number of published works of each one, and the order has been established according to this parameter. All changes were highlighted in red colour.

Reviewer 2 Report

Comments and Suggestions for Authors

This narrative review explores the role of oxidative stress in multiple sclerosis (MS) and evaluates the effectiveness of antioxidant treatments in both experimental models and clinical trials. As per the comprehensive literature cited, the antioxidants have shown promising results in improving clinical and biochemical parameters in animal models, however, the clinical trials conducted so far have been limited, with short-term studies and small patient groups providing inconclusive results. The authors suggest the need for long-term, randomized, multicenter clinical trials to better assess the potential of antioxidant therapies in MS treatment.

Overall, the manuscript provides a thorough overview of antioxidant treatments in the context of multiple sclerosis (MS), covering both experimental models and clinical trials. The focus on oxidative stress and antioxidant treatments in MS is highly relevant, given the ongoing research into potential therapeutic strategies for this chronic disease. The manuscript acknowledges the limitations of existing clinical trials, emphasizing the need for more robust and long-term studies. This critical perspective is important for guiding future research efforts.

The following are my concerns about the manuscript that need to be addressed before being considered for publication in Biomolecules journals

1.      The article submitted by Félix Javier Jiménez-Jiménez and colleagues resembles a literature review chapter of a PhD thesis more than a focused review paper on the therapeutic uses of antioxidants against multiple sclerosis. While a thesis literature review aims to provide a comprehensive background and justify the research, a review article should be more concise, thematic, and critically engage with the most relevant studies to advance the field. This article's broad scope, extensive detail, and lack of a focused narrative make it more suited to a thesis context rather than the concise, synthesis-driven format expected in a peer-reviewed journal review. To improve, the article should shift from summarizing each study individually to abstracting and synthesizing the outcomes from similar studies, highlighting key trends and gaps, and offering new insights or future research directions.

2.      The review provides a good overview of study outcomes but could be strengthened by a deeper discussion of the underlying mechanisms by which oxidative stress and antioxidants influence MS pathogenesis. This would enhance the scientific depth of the review.

3.      Many sentences in the manuscript are very long, complex, and difficult for readers to understand. It is advisable to break these long complex sentences into short and simple sentence for better understanding of the broader readership.

4.      Several terms are hyperlinked, throughout the manuscripts, such as in Lines 226, 436, 437, 481, 482, 505, 506, 680, 681, 682, 717, and 733. These terms are hyperlinked to Wikipedia in most cases and it seems that these terms have been directly copied from there. Please remove these hyperlinks.

5.      There are several abbreviations used in the manuscript that are not elaborated. Please elaborate all the abbreviations when they appear first in the manuscript.

6.      There is inconsistency in format at various places such as N-acetyl-cysteine in line 688 and N-acetylcysteine in line 686. Please write all terminology in a uniform format throughout the manuscript.

7.      Write all the genes' names in italic font throughout the manuscript.

8.      No antioxidant name has been mentioned in lines 114 to 117 and the reference cited in these lines is about thoctic acid, not ALA.

9.      Line 164, change “of” to “or” in “alone of associated”.

10.  Line, 197, add “it” before “can” in “In addition, can decrease”.

11.  Line 200, replace “and” with “an” in “causes and increase”.

12.  Line 421, correct the spelling of “Vegetable oil.

13.  Correct the formula of Hydrogen peroxide in lines 438 and 729 to H2O2

14.  Correct the spelling of tocopherol in line 545.

15.  Line 660, “Administration of the iNOS inhibitors aminoguanidine” iNOS is the abbreviated form of inhibitor of Nitric oxide synthase, therefore, delete “inhibitors”.

16.  Line 691, delete “amide”.

17.  Line 691, “C3H.SW/C57/bL” should be C3H.SW/C57BL.

18.  Line 713, correct the spelling of with.

19.  743, rectify “an experimental model of EAE by is to inhibit the activation of dendritic cells”.

20.  Line 747, change “citric” to “citrus”

21.  Line 763, rectify “the chemokine MCP-1 from in primary murine microglia”.

22.  Line 783, change “Acetyl-l-carnitine” to “acetyl-L-carnitine”.

23.  Line 970, “Biomedicine” shouldn’t be in bold font.

24.  Line 975, volume and issue number shouldn’t be bold.

25.  Line 983, Correct and complete the reference.

26.  Line 1235, delete duplicated 2013.

27.  Line 1477, protect shouldn’t be bold.

28.  Line 1478, delete duplicated 2020.

29.  Line 1626, “Naunyn” should be in italic font.

Regards 

Comments on the Quality of English Language

The quality of the English language is fine. However, many long and complex sentences have been used. It is advisable to break long complex sentences into short and simple sentences for a better understanding of readers from diverse backgrounds. Additionally, several typographic and spelling mistakes have been pointed out in the comments section, which need rectification. 

Author Response

The following are my concerns about the manuscript that need to be addressed before being considered for publication in Biomolecules journals

  1. The article submitted by Félix Javier Jiménez-Jiménez and colleagues resembles a literature review chapter of a PhD thesis more than a focused review paper on the therapeutic uses of antioxidants against multiple sclerosis. While a thesis literature review aims to provide a comprehensive background and justify the research, a review article should be more concise, thematic, and critically engage with the most relevant studies to advance the field. This article's broad scope, extensive detail, and lack of a focused narrative make it more suited to a thesis context rather than the concise, synthesis-driven format expected in a peer-reviewed journal review. To improve, the article should shift from summarizing each study individually to abstracting and synthesizing the outcomes from similar studies, highlighting key trends and gaps, and offering new insights or future research directions.

The purpose of this article, as discussed in the introduction, was to provide a comprehensive and broad review of the state of the subject. We have added a comment summarizing the results of the studies of each of the antioxidants in experimental models or patients with MS in each of the sections.

  1. The review provides a good overview of study outcomes but could be strengthened by a deeper discussion of the underlying mechanisms by which oxidative stress and antioxidants influence MS pathogenesis. This would enhance the scientific depth of the review.

OK, we have added a new paragraph in the discussion together with a figure related to the action of antioxidants in the pathogenetic mechanisms of MS.

  1. Many sentences in the manuscript are very long, complex, and difficult for readers to understand. It is advisable to break these long complex sentences into short and simple sentence for better understanding of the broader readership. OK, we have revised exhaustively the manuscript and rewritten complex sentences.
  2. Several terms are hyperlinked, throughout the manuscripts, such as in Lines 226, 436, 437, 481, 482, 505, 506, 680, 681, 682, 717, and 733. These terms are hyperlinked to Wikipedia in most cases and it seems that these terms have been directly copied from there. Please remove these hyperlinks. OK, all removed
  3. There are several abbreviations used in the manuscript that are not elaborated. Please elaborate all the abbreviations when they appear first in the manuscript. OK, all revised exhaustively and added
  4. There is inconsistency in format at various places such as N-acetyl-cysteine in line 688 and N-acetylcysteine in line 686. Please write all terminology in a uniform format throughout the manuscript. OK, corrected throughout the text
  5. Write all the genes' names in italic font throughout the manuscript. OK, all changed throughout the manuscript
  6. No antioxidant name has been mentioned in lines 114 to 117 and the reference cited in these lines is about thoctic acid, not ALA. OK, corrected. ALA is also known as thioctic acid.
  7. Line 164, change “of” to “or” in “alone of associated”. Ok. Done
  8. Line, 197, add “it” before “can” in “In addition, can decrease”. Ok. Done
  9. Line 200, replace “and” with “an” in “causes and increase”. Ok. Done
  10. Line 421, correct the spelling of “Vegetable oil. Ok. Done
  11. Correct the formula of Hydrogen peroxide in lines 438 and 729 to H2O2 Ok. Done
  12. Correct the spelling of tocopherol in line 545. Ok. Done
  13. Line 660, “Administration of the iNOS inhibitors aminoguanidine” iNOS is the abbreviated form of inhibitor of Nitric oxide synthase, therefore, delete “inhibitors”. iNOS is the abbreviation of inducible nitric oxide synthase, therefore the term iNOS inhibitors is OK, We have added the definition of this abbreviation. 
  14. Line 691, delete “amide”. OK, Deleted
  15. Line 691, “C3H.SW/C57/bL” should be C3H.SW/C57BL. OK, corrected
  16. Line 713, correct the spelling of with. Ok, corrected
  17. 743, rectify “an experimental model of EAE by is to inhibit the activation of dendritic cells”. OK, corrected.
  18. Line 747, change “citric” to “citrus” OK, corrected.
  19. Line 763, rectify “the chemokine MCP-1 from in primary murine microglia”. OK, corrected.
  20. Line 783, change “Acetyl-l-carnitine” to “acetyl-L-carnitine”. OK, corrected.
  21. Line 970, “Biomedicine” shouldn’t be in bold font. OK, corrected.
  22. Line 975, volume and issue number shouldn’t be bold. OK, corrected.
  23. Line 983, Correct and complete the reference. This reference corresponds to an e-book chapter. It is cited according to PubMed.
  24. Line 1235, delete duplicated 2013. This reference is correct according to Pubmed. In this journal, the volume number is the same as the year.
  25. Line 1477, protect shouldn’t be bold. OK, corrected.
  26. Line 1478, delete duplicated 2020. This reference is correct according to Pubmed. In this journal, the volume number is the same as the year.
  27. Line 1626, “Naunyn” should be in italic font. OK, corrected.

All changes were highlighted in red colour.

Reviewer 3 Report

Comments and Suggestions for Authors

I thank the authors for their manuscript. They have done a great job of analyzing a large volume of literature data. It is assumed that oxidative stress plays a certain role in the pathogenesis of multiple sclerosis. Therefore, treatment with antioxidant drugs may be important in the complex therapy of this disease. The authors reviewed the action of ALA, melatonin, EGCG, curcumin, resveratrol, pentoxifylline, vegetable and animal oils, CoQ10, antioxidant vitamins, uric acid, bilirubin, NOS inhibitors, scavengers and precursors of NO, N-acetylcysteine, flavonoid compounds, and other molecules with antioxidant action. Many antioxidant substances have demonstrated beneficial effects in experimental models of MS and may find application in the therapy of this disease.
The authors have reviewed all required sections. This review systematizes a large number of experimental studies in this area. The authors grouped all studies according to the object of study: (1) cell culture, (2) experimental models of MS, especially in experimental autoimmune encephalomyelitis (EAE) and in the cuprizone-induced demyelination model, (3) clinical trials in patients diagnosed with multiple sclerosis. This is a good decision.
The conclusions are consistent with the data presented. The authors emphasize that some studies were evaluated in the short term or with a small sample of patients. References are appropriate. The authors could supplement their review with a figure that would illustrate the data described.
I believe the manuscript can be accepted for publication.

Author Response

I thank the authors for their manuscript. They have done a great job of analyzing a large volume of literature data. It is assumed that oxidative stress plays a certain role in the pathogenesis of multiple sclerosis. Therefore, treatment with antioxidant drugs may be important in the complex therapy of this disease. The authors reviewed the action of ALA, melatonin, EGCG, curcumin, resveratrol, pentoxifylline, vegetable and animal oils, CoQ10, antioxidant vitamins, uric acid, bilirubin, NOS inhibitors, scavengers and precursors of NO, N-acetylcysteine, flavonoid compounds, and other molecules with antioxidant action. Many antioxidant substances have demonstrated beneficial effects in experimental models of MS and may find application in the therapy of this disease.
The authors have reviewed all required sections. This review systematizes a large number of experimental studies in this area. The authors grouped all studies according to the object of study: (1) cell culture, (2) experimental models of MS, especially in experimental autoimmune encephalomyelitis (EAE) and in the cuprizone-induced demyelination model, (3) clinical trials in patients diagnosed with multiple sclerosis. This is a good decision.
The conclusions are consistent with the data presented. The authors emphasize that some studies were evaluated in the short term or with a small sample of patients. References are appropriate. The authors could supplement their review with a figure that would illustrate the data described.
I believe the manuscript can be accepted for publication.

OK, we have added a new paragraph in the discussion together with a figure related with the action of antioxidants in the pathogenetic mechanisms of MS. All changes were highlighted in red colour.

Round 2

Reviewer 2 Report

Comments and Suggestions for Authors

Thank you for addressing all the points raised. I am concerned about this article's high similarity index, which is 48%.

Author Response

We have now revised exhaustively the text. Many of the similarity is due to the references (when an article is cited in another, the reference has already been cited at least once, and similarity is inevitable) chemical names of products or abbreviations (many of which are standard). In the rest of the similarities, the pertinent changes have been made